# BAYESIAN VARIATIONAL AUTOENCODERS FOR UNSUPERVISED OUT-OF-DISTRIBUTION DETECTION

## ABSTRACT

Despite their successes, deep neural networks still make unreliable predictions when faced with test data drawn from a distribution different to that of the training data, constituting a major problem for AI safety. While this motivated a recent surge in interest in developing methods to detect such out-of-distribution (OoD) inputs, a robust solution is still lacking. We propose a new probabilistic, unsupervised approach to this problem based on a Bayesian variational autoencoder model, which estimates a full posterior distribution over the decoder parameters using stochastic gradient Markov chain Monte Carlo, instead of fitting a point estimate. We describe how information-theoretic measures based on this posterior can then be used to detect OoD data both in input space as well as in the model's latent space. The effectiveness of our approach is empirically demonstrated.

## 1 INTRODUCTION

**Outlier detection in input space.** While deep neural networks (DNNs) have successfully tackled complex real-world problems in various domains including vision, speech and language (Schmidhuber, 2015; LeCun et al., 2015; Goodfellow et al., 2016), they still face significant limitations that make them unfit for safety-critical applications (Amodei et al., 2016). One well-known shortcoming of DNNs is that when faced with test data points coming from a different distribution than the data the network was exposed to during training, the DNN will not only output wrong predictions, but it will do so with high confidence. The lack of robustness of DNNs to such *out-of-distribution* (OoD) inputs (or *outliers/anomalies*) was recently addressed by the development of various methods to detect OoD inputs. Most existing OoD detection approaches are *task-specific* in that they consider a certain prediction task (typically classification) and use the corresponding output labels for the supervised training of a deep discriminative model to produce the desired target output as well as some confidence score (Hendrycks & Gimpel, 2016; Hendrycks et al., 2018; Liang et al., 2017). Alternatively, *task-agnostic* OoD detection methods solely use the input data for the unsupervised training of a deep generative model (DGM). One simple and seemingly sensible approach to detect a potential OoD input $\mathbf{x}^*$ is to train a likelihood-based DGM (e.g. a VAE, auto-regressive DGM, or flow-based DGM) by (approximately) maximizing the likelihood $p(\mathbb{D}|\theta)$ of the model parameters $\theta$ under the training data $\mathbb{D}$, and to then estimate the probability $p(\mathbf{x}^*|\theta)$ that $\mathbf{x}^*$ was generated by the model $\theta$ (Bishop, 1994): If $p(\mathbf{x}^*|\theta)$ is large, then $\mathbf{x}^*$ must be in-distribution, and OoD otherwise. However, recent works have shown that this likelihood-based approach does not work in general, as deep generative models sometimes assign *higher* probability to OoD data than to in-distribution data (Nalisnick et al., 2018; Choi & Jang, 2018). Motivated by this finding, recent works have tried to develop more effective scores by correcting the likelihood estimate (Choi & Jang, 2018; Ren et al., 2019; Nalisnick et al., 2019). However, while these approaches improve upon the likelihood score, we argue that OoD detection scores which are fundamentally based on the likelihood values are not robust, as the likelihood estimates produced by commonly used DGMs are not reliable for OoD data, similar to how deep discriminative models produce unreliable output predictions on OoD data.

**Outlier detection in latent space.** In a distinct line of research, recent works have tackled the challenge of efficiently optimizing a complex black-box function $f : \mathbb{X} \to \mathbb{R}, f(\mathbf{x}) = y$ defined over high-dimensional, richly structured input domains $\mathbb{X}$ (e.g. graphs, images or text). Given examples $\mathbb{D} = \{(\mathbf{x}_i, y_i)\}_{i=1}^N$, these methods jointly train a VAE on inputs $\mathbf{x}$ and a predictive model $g : \mathbb{Z} \to \mathbb{R}, g(\mathbf{z}) = y$ mapping from latent codes $\mathbf{z}$ to targets $y$, to then perform the optimization

w.r.t. $y$ in the low-dimensional, continuous latent space $\mathbb{Z}$ instead of in input space $\mathbb{X}$ (Gómez-Bombarelli et al., 2018). While these approaches have achieved state-of-the-art results in important domains including automatic chemical design and automatic machine learning (Gómez-Bombarelli et al., 2018; Luo et al., 2018; Lu et al., 2018), their practical effectiveness is limited by their ability to handle the following trade-off: They need to find inputs $\mathbf{x}$ that both have a high target value $y$ and are sufficiently novel (i.e., not too close to inputs in the training data $\mathbb{D}$), while at the same time ensuring that the optimization w.r.t. $y$ does not progress into regions of the latent space $\mathbb{Z}$ too far away from the training data, which might yield latent points $\mathbf{z}$ that decode to semantically meaningless and/or syntactically invalid inputs $\mathbf{x}$ (Kusner et al., 2017; Griffiths & Hernández-Lobato, 2017; Brookes et al., 2019; Mahmood & Hernández-Lobato, 2019; Alperstein et al., 2019). We observe that the required ability to quantify the semantic/syntactic distance of latent codes $\mathbf{z}$ to the training data directly corresponds to the ability to detect outliers in latent space $\mathbb{Z}$.

**Our approach.** In this work, we propose a principled, unsupervised, probabilistic method to simultaneously tackle the challenge of detecting outliers $\mathbf{x}^*$ in input space $\mathbb{X}$ (without directly relying on likelihood estimates) as well as outliers $\mathbf{z}^*$ in latent space $\mathbb{Z}$. To this end, we take an information-theoretic perspective on OoD detection, and propose to use the (expected) *informativeness* of an input $\mathbf{x}^*$ / latent $\mathbf{z}^*$ as a proxy for whether $\mathbf{x}^*$ / $\mathbf{z}^*$ is OoD or not. To quantify this informativeness, we take inspiration from information-theoretic active learning (MacKay, 1992) and leverage probabilistic inference techniques to maintain a posterior distribution over the parameters of a deep generative model, in particular of a variational autoencoder (VAE) (Kingma & Welling, 2013; Rezende et al., 2014). This results in a *Bayesian VAE* (BVAE) model, where instead of fitting a point estimate of the decoder parameters via maximum likelihood, we estimate their posterior distribution using samples generated via stochastic gradient Markov chain Monte Carlo (MCMC). The informativeness of an unobserved input $\mathbf{x}^*$ / latent $\mathbf{z}^*$ is then quantified by measuring the (expected) change in the posterior over model parameters after having observed $\mathbf{x}^*$ / $\mathbf{z}^*$. In summary, our contributions are as follows: (a) We propose a Bayesian VAE model which uses state-of-the-art Bayesian inference techniques to estimate a posterior distribution over the decoder parameters (Section 3.2). (b) We describe how this model can be used to detect outliers both in input space (Section 3.3) as well as in the model's latent space (Section 3.4) using information-theoretic metrics. (c) We empirically demonstrate that our approach outperforms state-of-the-art outlier detection methods (Section 5).

## 2 BACKGROUND

### 2.1 VARIATIONAL AUTOENCODERS

Consider a latent-variable model $p(\mathbf{x}, \mathbf{z}|\theta)$ with marginal likelihood $p(\mathbf{x}|\theta) = \int p(\mathbf{x}, \mathbf{z}|\theta)d\mathbf{z}$, where $\mathbf{x}$ are observed variables, $\mathbf{z}$ are local latent variables (i.e., individually tied to some $\mathbf{x}$), and $\theta$ are global latent variables (i.e., shared among all $\mathbf{x}$).[1] For clarity, we will mostly refer to $\mathbf{z}$ simply as *latent variables* (i.e., without the prefix *local*), and to $\theta$ as *model/decoder parameters*. We assume that $p(\mathbf{x}, \mathbf{z}|\theta) = p(\mathbf{x}|\mathbf{z}, \theta)p(\mathbf{z})$, i.e., the joint density $p(\mathbf{x}, \mathbf{z}|\theta)$ factorizes into a prior distribution $p(\mathbf{z})$ on the local latent variables $\mathbf{z}$ and a parameterized likelihood $p(\mathbf{x}|\mathbf{z}, \theta)$ of the observed variables $\mathbf{x}$ given $\mathbf{z}$. We furthermore assume the $\mathbf{z}$ to be continuous, such that computing $p(\mathbf{x}|\theta)$ in generally intractable. Such a model is called a *variational autoencoder* (VAE) (Kingma & Welling, 2013; Rezende et al., 2014) if $\theta$ are the parameters of a deep neural network, and the corresponding intractable posterior distribution $p(\mathbf{z}|\mathbf{x}, \theta)$ over the latent variables $\mathbf{z}$ is approximated using amortized variational inference via another deep neural network $q(\mathbf{z}|\mathbf{x}, \phi)$ parameterized by $\phi$ (typically called the encoder, inference network or recognition network). Given some dataset $\mathbb{D} = \{\mathbf{x}_i\}_{i=1}^N$ of $N$ input vectors $\mathbf{x}_i \in \mathbb{X}$ drawn i.i.d. from some underlying data distribution $p^*(\mathbf{x})$, the parameters $\theta$ and $\phi$ of a VAE are then learned by maximizing the evidence lower bound (ELBO) $\mathcal{L}_{\theta,\phi}$, which is a lower bound to the marginal log-likelihood $\log p(\mathbf{x}|\theta)$, i.e.,

$$\mathcal{L}_{\theta,\phi}(\mathbf{x}) = \mathbb{E}_{q(\mathbf{z}|\mathbf{x},\phi)}[\log p(\mathbf{x}|\mathbf{z}, \theta)] - D_{\mathrm{KL}}(q(\mathbf{z}|\mathbf{x}, \phi)\|p(\mathbf{z})) \quad \leq \quad \log p(\mathbf{x}|\theta) \quad (1)$$

for some input $\mathbf{x} \in \mathbb{D}$. By independence, $\mathcal{L}_{\theta,\phi}(\mathbb{D}) = \sum_{\mathbf{x}\in\mathbb{D}} \mathcal{L}_{\theta,\phi}(\mathbf{x})$. Since maximizing the ELBO approximately maximizes $\log p(\mathbb{D}|\theta)$, this training procedure can be viewed as approximate max-

---

[1] Note that most literature on VAEs includes the parameters $\theta$ as a subscript, e.g., $p_\theta(\mathbf{x}, \mathbf{z})$, assuming a fixed point estimate $\theta_{\mathrm{MLE}}$ of $\theta$, i.e., $p_{\theta_{\mathrm{MLE}}}(\mathbf{x}, \mathbf{z}) = p(\mathbf{x}, \mathbf{z}|\theta = \theta_{\mathrm{MLE}})$. We instead use the notation $p(\mathbf{x}, \mathbf{z}|\theta)$ to make explicit that we assume $\theta$ to be a random variable governed by some distribution (to be detailed in Section 3).

imum likelihood estimation (MLE). In practice, $\mathcal{L}_{\theta,\phi}(\mathbf{x})$ in Eq. (1) is maximized by mini-batch stochastic gradient-based optimization using low-variance, unbiased, stochastic Monte Carlo estimators of $\nabla \mathcal{L}_{\theta,\phi}$ obtained via the reparametrization trick. Finally, one can approximate the probability $p(\mathbf{x}|\theta,\phi)$ of a data point $\mathbf{x}$ under the generative model via importance sampling w.r.t. the variational posterior $q(\mathbf{z}|\mathbf{x},\phi)$, i.e.,

$$p(\mathbf{x}|\theta,\phi) = \mathbb{E}_{q(\mathbf{z}|\mathbf{x},\phi)}\left[\frac{p(\mathbf{x}|\mathbf{z},\theta)p(\mathbf{z})}{q(\mathbf{z}|\mathbf{x},\phi)}\right] \simeq \frac{1}{K}\sum_{k=1}^{K}\frac{p(\mathbf{x}|\mathbf{z}_k,\theta)p(\mathbf{z}_k)}{q(\mathbf{z}_k|\mathbf{x},\phi)}, \quad \mathbf{z}_k \sim q(\mathbf{z}|\mathbf{x},\phi) . \quad (2)$$

Note that the likelihood $p(\mathbf{x}|\theta,\phi)$ in Eq. (2) is conditioned on both $\theta$ and $\phi$, to make explicit the dependence on the parameters $\phi$ of the proposal distribution $q(\mathbf{z}|\mathbf{x},\phi)$.

## 2.2 BAYESIAN LEARNING IN NEURAL NETWORKS VIA STOCHASTIC GRADIENT MCMC

To generate samples $\theta \sim p(\theta|\mathbb{D})$ of parameters $\theta$ of a deep neural network, one can use state-of-the-art stochastic gradient Markov chain Monte Carlo methods such as stochastic gradient Hamiltonian Monte Carlo (SGHMC). Hamiltonian Monte Carlo (HMC) (Duane et al., 1987; Betancourt, 2017) is a method for generating samples $\theta \sim p(\theta|\mathbb{D})$ in a Metropolis-Hastings framework that efficiently explores the state space. In particular, consider the posterior distribution $p(\theta|\mathbb{D}) \propto \exp(-U(\theta,\mathbb{D}))$ with potential energy function $U(\theta,\mathbb{D}) = -\log p(\mathbb{D},\theta) = -\log(p(\mathbb{D}|\theta)p(\theta)) = -\sum_{\mathbf{x}\in\mathbb{D}}\log p(\mathbf{x}|\theta) - \log p(\theta)$ induced by the prior $p(\theta)$ and marginal log-likelihood $\log p(\mathbf{x}|\theta)$. HMC generates samples $\theta \sim p(\theta|\mathbb{D})$ by simulating Hamiltonian dynamics, which involves evaluating the gradient $\nabla_\theta U(\theta)$ of $U$. However, direct computation of this gradient requires examination of the entire dataset $\mathbb{D}$ (due to the summation of the log-likelihood over all $\mathbf{x} \in \mathbb{D}$), which might become prohibitively costly for large datasets. To tackle this issue, Chen et al. (2014) recently proposed a scalable HMC variant called stochastic gradient Hamiltonian Monte Carlo (SGHMC), which considers a noisy, unbiased estimate of the gradient computed from a minibatch $\mathbb{M}$ of points sampled uniformly at random from $\mathbb{D}$ (i.e., akin to minibatch-based optimization algorithms such as variants of stochastic gradient descent), i.e.,

$$\nabla_\theta U(\theta,\mathbb{D}) \simeq \nabla_\theta U(\theta,\mathbb{M}) = -\frac{|\mathbb{D}|}{|\mathbb{M}|}\sum_{\mathbf{x}\in\mathbb{M}}\nabla_\theta \log p(\mathbf{x}|\theta) - \nabla_\theta \log p(\theta) . \quad (3)$$

# 3 BAYESIAN VARIATIONAL AUTOENCODERS FOR OOD DETECTION

## 3.1 MOTIVATION AND INTUITION

We take the following information-theoretic perspective on OoD detection. Consider an active learning scenario, where we have computed an estimate of our model parameters $\theta$ based on some observations $\mathbb{D}$, and want to add an unobserved input $\mathbf{x}^*$ to the training set $\mathbb{D}$ to improve our estimate of $\theta$ as much as possible. To this end, in information-theoretic active learning (MacKay, 1992), it is observed that every potential input $\mathbf{x}^*$ contains a certain amount of information about the values of the model parameters. In other words, every data point $\mathbf{x}^*$ helps us to some extent in updating our model parameters to find the optimal ones which capture the true underlying data-generating process. Now, we argue that inputs which are uninformative about the model parameters are likely similar to the data points already in the training set $\mathbb{D}$, i.e., they are likely in-distribution inputs. In contrast, inputs which are very informative about the model parameters are likely different from everything we have seen so far in the training data $\mathbb{D}$, i.e., they are likely OoD inputs. We thus propose to use the *informativeness* of a datum $\mathbf{x}^*$ as a proxy for whether $\mathbf{x}^*$ is OoD or not. To quantify this informativeness, information-theoretic active learning approaches leverage probabilistic inference techniques to maintain a posterior distribution $p(\theta|\mathbb{D})$ over model parameters $\theta$ given data $\mathbb{D}$ (i.e., instead of fitting a point estimate of $\theta$ via maximum likelihood estimation). Given this posterior $p(\theta|\mathbb{D})$, the informativeness of an unobserved datum $\mathbf{x}^*$ is then quantified by measuring the (expected) change in the posterior after having observed $\mathbf{x}^*$, i.e., the change required to update $p(\theta|\mathbb{D})$ to the posterior $p(\theta|\mathbb{D} \cup \{\mathbf{x}^*\})$. We follow this approach and propose to use a *Bayesian VAE* (BVAE) model, where instead of fitting a point estimate of the parameters via maximum likelihood, we estimate their posterior distribution using samples generated via stochastic gradient MCMC.

## 3.2 The Bayesian VAE (BVAE)

In contrast to an ordinary VAE, where we fit the generative model parameters $\theta$ via (approximate) MLE, i.e., $\theta_{\text{MLE}} = \arg\max_\theta \mathcal{L}(\mathbb{D})_{\theta,\phi}$, to obtain the likelihood $p(\mathbf{x}|\mathbf{z}, \theta_{\text{MLE}})$, we place a prior $p(\theta)$ over $\theta$ and estimate its full posterior distribution $p(\theta|\mathbb{D}) \propto p(\mathbb{D}|\theta)p(\theta)$, to induce the generator likelihood $p(\mathbf{x}|\mathbf{z}, \mathbb{D}) = \int p(\mathbf{x}|\mathbf{z}, \theta)p(\theta|\mathbb{D})d\theta = \mathbb{E}_{p(\theta|\mathbb{D})}[p(\mathbf{x}|\mathbf{z}, \theta)]$. This induces the marginal likelihood

$$p(\mathbf{x}|\mathbb{D}) = \mathbb{E}_{p(\mathbf{z})}[p(\mathbf{x}|\mathbf{z}, \mathbb{D})] = \int \int p(\mathbf{x}|\mathbf{z}, \theta)p(\theta|\mathbb{D})p(\mathbf{z})d\theta d\mathbf{z} = \mathbb{E}_{p(\theta|\mathbb{D})}[p(\mathbf{x}|\theta)] \quad (4)$$

which marginalizes over *both* the local latent variables $\mathbf{z}$ and the global latent variables $\theta$. The generative process defined in Eq. (4) first draws a latent vector $\mathbf{z} \sim p(\mathbf{z})$ from its prior and a decoder parameterization $\theta \sim p(\theta|\mathbb{D})$ from its posterior, and then generates $\mathbf{x} \sim p(\mathbf{x}|\mathbf{z}, \theta)$ by passing both through the likelihood. Training this model now involves performing Bayesian inference over *both* $\mathbf{z}$ and $\theta$, resulting in the posterior $p(\mathbf{z}|\mathbf{x}, \mathbb{D})$ over $\mathbf{z}$ and the posterior $p(\theta|\mathbb{D}) \propto p(\mathbb{D}|\theta)p(\theta)$ over $\theta$. Since both posteriors are intractable for the model we consider, we will now discuss how to approximate inference over $\mathbf{z}$ and $\theta$, respectively.

### 3.2.1 Inference over the latent variables $\mathbf{z}$

To estimate the posterior $p(\mathbf{z}|\mathbf{x}, \mathbb{D})$ over $\mathbf{z}$, we follow an ordinary VAE and resort to amortized variational inference via a recognition network $q(\mathbf{z}|\mathbf{x}, \phi)$ with variational parameters $\phi$, using the ELBO $\mathcal{L}_{\theta,\phi}(\mathbf{x}) \leq \log p(\mathbf{x}|\theta)$ in Eq. (1). We learn the encoder parameters $\phi$ via approximate MLE, using a low-variance, unbiased Monte Carlo estimator of the stochastic gradient $\nabla_\phi \mathcal{L}_{\theta,\phi}(\mathbf{x})$, i.e.,

$$\nabla_\phi \mathcal{L}_{\theta,\phi}(\mathbf{x}) \simeq \frac{1}{L}\sum_{l=1}^{L} \nabla_\phi \log\left[p(\mathbf{x}|\mathbf{z}_l, \theta)\right] - \nabla_\phi D_{\text{KL}}(q(\mathbf{z}|\mathbf{x}, \phi)\|p(\mathbf{z})), \quad \mathbf{z}_l \sim q(\mathbf{z}|\mathbf{x}, \phi) \quad (5)$$

where the posterior samples $\mathbf{z}_l \sim q(\mathbf{z}|\mathbf{x}, \phi)$ are generated using the reparametrization trick.

### 3.2.2 Inference over the model parameters $\theta$

To generate posterior samples $\theta \sim p(\theta|\mathbb{D})$ of decoder parameters (e.g. as required to approximate the gradient $\nabla_\phi \mathcal{L}_\phi(\mathbf{x})$ in Eq. (5)), we propose to use SGHMC (cf. Section 2). However, the gradient of the energy function $\nabla_\theta U(\theta, \mathbb{M})$ in Eq. (3) used for simulating the Hamiltonian dynamics requires us to evaluate the log-likelihood $\log p(\mathbf{x}|\theta)$, which is intractable in a VAE and thus also in our model. To alleviate this issue, we approximate the log-likelihood appearing in $\nabla_\theta U(\theta, \mathbb{M})$ by the ELBO $\mathcal{L}_{\theta,\phi}(\mathbf{x})$ of an ordinary VAE in Eq. (1). Given a set $\Theta = \{\theta_m\}_{m=1}^{M}$ of posterior samples $\theta_m \sim p(\theta|\mathbb{D})$, we can more intuitively think of working with a finite mixture/ensemble of decoders/generative models $p(\mathbf{x}|\mathbf{z}, \mathbb{D}) = \mathbb{E}_{p(\theta|\mathbb{D})}[p(\mathbf{x}|\mathbf{z}, \theta)] \simeq \frac{1}{M}\sum_{\theta\in\Theta} p(\mathbf{x}|\mathbf{z}, \theta)$ (cf. Eq. (4)).

### 3.2.3 Inference over the variational parameters $\phi$

Recall that the goal of amortized variational inference is to *learn* how to do posterior inference, by finding the optimal parameters $\phi_{\text{MLE}} = \arg\max_\phi \mathcal{L}(\mathbb{D})_{\theta,\phi}$ (cf. Eq. (1)) of an inference network $i : \mathbb{X} \to \Psi : i_\phi(\mathbf{x}) = \psi$ mapping inputs $\mathbf{x}$ to parameters $\psi$ of the variational posterior $q_\psi(\mathbf{z}) = q(\mathbf{z}|\mathbf{x}, \phi)$ over $\mathbf{z}$. However, one fundamental shortcoming of fitting an inference network via MLE is that the model $q(\mathbf{z}|\mathbf{x}, \phi_{\text{MLE}})$ will not generalize to OoD inputs, but instead produce confidently wrong posterior inferences for OoD inputs. To alleviate this issue, we refrain from fitting a point estimate $\phi_{\text{MLE}}$, and instead consider a Bayesian treatment of the variational parameters $\phi$ (i.e., in addition to the Bayesian treatment of the decoder parameters $\theta$ discussed earlier). While this might seem strange conceptually, it allows us to quantify our epistemic uncertainty in the amortized inference of $\mathbf{z}$, intuitively increasing the flexibility of the recognition network. We thus also place a prior $p(\phi)$ over $\phi$ and infer the posterior $p(\phi|\mathbb{D}) \propto p(\mathbb{D}|\phi)p(\phi)$, yielding the amortized posterior

$$q(\mathbf{z}|\mathbf{x}, \mathbb{D}) = \int q(\mathbf{z}|\mathbf{x}, \phi)p(\phi|\mathbb{D})d\phi = \mathbb{E}_{p(\phi|\mathbb{D})}[q(\mathbf{z}|\mathbf{x}, \phi)] \simeq \frac{1}{M}\sum_{j=1}^{M} q(\mathbf{z}|\mathbf{x}, \phi_j), \ \phi_j \sim p(\phi|\mathbb{D}) . \quad (6)$$

We also use SGHMC to generate the posterior samples $\phi_j \sim p(\phi|\mathbb{D})$, again approximating the log-likelihood $\log p(\mathbf{x}|\phi, \mathbb{D})$ in $\nabla_\phi U(\phi, \mathbb{M})$ (cf. Eq. (3)) by the ELBO $\mathcal{L}_{\theta,\phi}(\mathbf{x})$ in Eq. (1). Given a set $\Phi = \{\phi_j\}_{j=1}^{M}$ of posterior samples $\phi_j \sim p(\phi|\mathbb{D})$, we can again more intuitively think of Eq. (6) as defining a finite mixture/ensemble of inference networks $q(\mathbf{z}|\mathbf{x}, \mathbb{D}) \simeq \frac{1}{M}\sum_{\phi\in\Phi} q(\mathbf{z}|\mathbf{x}, \phi)$.

**Inference over $\theta$ revisited.** When doing inference over both $\phi$ and $\theta$, we need to slightly adapt the SGHMC sampling procedure for the decoder parameters $\theta \sim p(\theta|\mathbb{D})$ described in Section 3.2.2. In particular, instead of using the ELBO $\mathcal{L}_{\theta,\phi}(\mathbf{x})$ of an ordinary VAE in Eq. (1) (which is based on the recognition network $q(\mathbf{z}|\mathbf{x}, \phi)$ and thus depends on both $\theta$ and $\phi$), we now need to use the ELBO

$$\mathcal{L}_\theta(\mathbf{x}) = \mathbb{E}_{q(\mathbf{z}|\mathbf{x},\mathbb{D})}[\log p(\mathbf{x}|\mathbf{z}, \theta)] - D_{\mathrm{KL}}(q(\mathbf{z}|\mathbf{x}, \mathbb{D})\|p(\mathbf{z})) \quad \leq \quad \log p(\mathbf{x}|\theta, \mathbb{D}) \qquad (7)$$

which depends on $\theta$ only, as it is based on the encoder mixture $q(\mathbf{z}|\mathbf{x}, \mathbb{D})$ in Eq. (6) and thus integrates the variational parameters $\phi$ over their posterior $p(\phi|\mathbb{D})$. We then obtain an approximation to the stochastic gradient $\nabla_\theta U(\theta, \mathbb{M})$ in Eq. (3) by using a Monte Carlo approximation of the gradient $\nabla_\theta \mathcal{L}_\theta(\mathbf{x})$ of the ELBO in Eq. (7), i.e.,

$$\nabla_\theta \mathcal{L}_\theta(\mathbf{x}) = \mathbb{E}_{q(\mathbf{z}|\mathbf{x},\mathbb{D})}[\nabla_\theta \log p(\mathbf{x}|\mathbf{z}, \theta)] \simeq \tfrac{1}{L} \sum_{l=1}^{L} \nabla_\theta \log p(\mathbf{x}|\mathbf{z}_l, \theta), \quad \mathbf{z}_l \sim q(\mathbf{z}|\mathbf{x}, \mathbb{D}) . \qquad (8)$$

The $\mathbf{z}_l \sim q(\mathbf{z}|\mathbf{x}, \mathbb{D})$ are sampled from the encoder mixture in Eq. (6) by first sampling the index of the mixture component uniformly at random, i.e., $j \sim \mathcal{U}[1, M]$ (due to the mixture weights all being $1/M$), and then sampling the latent vector $\mathbf{z}_l$ from the $j$-th mixture component, i.e., $\mathbf{z}_l \sim q(\mathbf{z}|\mathbf{x}, \phi_j)$.

### 3.3 DETECTING OUTLIERS IN INPUT SPACE

Training a BVAE (see Appendix B for pseudocode) yields the sets $\Phi = \{\phi_j\}_{j=1}^M$ and $\Theta = \{\theta_m\}_{m=1}^M$ of posterior samples $\phi_j \sim p(\phi|\mathbb{D})$ and $\theta_m \sim p(\theta|\mathbb{D})$ of encoder and decoder parameters, respectively. Given these samples, we follow ideas from information-theoretic active learning (MacKay, 1992) (cf. Section 1) and aim to detect if a certain test input $\mathbf{x}^*$ is OoD based on its informativeness about the model parameters $\theta$, as measured by the change in the posterior distribution over $\theta$ after having observed $\mathbf{x}^*$. In particular, assume that given some data $\mathbb{D}$ and prior $p(\theta)$, we have inferred the posterior $p(\theta|\mathbb{D}) = \frac{p(\mathbb{D}|\theta)p(\theta)}{\int p(\mathbb{D}|\theta)p(\theta)d\theta} = \frac{p(\mathbb{D}|\theta)}{p(\mathbb{D})}p(\theta)$ over $\theta$. In a sequential Bayesian setting, the posterior $p(\theta|\mathbb{D})$ then serves as the new prior, which, given a new observation $\mathbf{x}^*$, is updated to the posterior $p(\theta|\mathbb{D} \cup \{\mathbf{x}^*\}) = \frac{p(\mathbf{x}^*|\theta)p(\theta|\mathbb{D})}{\int p(\mathbf{x}^*|\theta)p(\theta|\mathbb{D})d\theta} = \frac{p(\mathbf{x}^*|\theta)}{p(\mathbf{x}^*|\mathbb{D})}p(\theta|\mathbb{D})$, i.e., by multiplying $p(\theta|\mathbb{D})$ by the normalized likelihood $\frac{p(\mathbf{x}^*|\theta)}{p(\mathbf{x}^*|\mathbb{D})}$. The intuition now is as follows: If $\mathbf{x}^*$ is very different from the previous observations in $\mathbb{D}$, then the updated posterior $p(\theta|\mathbb{D} \cup \{\mathbf{x}^*\})$ will likely be different from $p(\theta|\mathbb{D})$, to capture the atypicality of $\mathbf{x}^*$. In contrast, if $\mathbf{x}^*$ is very similar to the observations in $\mathbb{D}$, then the posterior $p(\theta|\mathbb{D} \cup \{\mathbf{x}^*\})$ will also be similar to the previous one $p(\theta|\mathbb{D})$. Thus, to detect whether $\mathbf{x}^*$ is OoD or not, we aim to quantify the change from $p(\theta|\mathbb{D})$ to $p(\theta|\mathbb{D} \cup \{\mathbf{x}^*\})$, which is captured by the factor $\frac{p(\mathbf{x}^*|\theta)}{p(\mathbf{x}^*|\mathbb{D})}$. Since in our case, the posterior $p(\theta|\mathbb{D})$ is represented by the finite set of samples $\Theta$, we find ourselves in a sequential Monte Carlo (or particle filtering) setting, such that for a given parameter/particle $\theta$ and input $\mathbf{x}^*$, the normalized likelihood $\frac{p(\mathbf{x}^*|\theta)}{p(\mathbf{x}^*|\mathbb{D})}$ can be written as

$$\frac{p(\mathbf{x}^*|\theta)}{p(\mathbf{x}^*|\mathbb{D})} \overset{(4)}{=} \frac{p(\mathbf{x}^*|\theta)}{\mathbb{E}_{p(\theta|\mathbb{D})}[p(\mathbf{x}^*|\theta)]} \simeq \frac{p(\mathbf{x}^*|\theta)}{\frac{1}{M}\sum_{\theta\in\Theta} p(\mathbf{x}^*|\theta)} = Mw_\theta, \quad \text{with} \quad w_\theta = \frac{p(\mathbf{x}^*|\theta)}{\sum_{\theta\in\Theta} p(\mathbf{x}^*|\theta)} \qquad (9)$$

where $w_\theta \in [0, 1]$ and $\sum_{\theta\in\Theta} w_\theta = 1$. The scalar $w_\theta$ defined in Eq. (9) can be interpreted as the probability that $\mathbf{x}^*$ was generated from the particle $\theta$, thus measuring how well $\mathbf{x}^*$ is explained by the model $\theta$ (and thus how useful $\theta$ is for describing the updated posterior $p(\theta|\mathbb{D}\cup\{\mathbf{x}^*\})$), relative to the other particles. More formally, the values $[w_\theta]_{\theta\in\Theta}$ correspond to the importance weights of the samples/particles $\theta \in \Theta$ drawn from the proposal distribution $p(\theta|\mathbb{D})$ (i.e., the previous posterior) for an importance sampling-based Monte Carlo approximation of an expectation w.r.t. the target distribution $p(\theta|\mathbb{D} \cup \{\mathbf{x}^*\})$ (i.e., the updated posterior after having observed $\mathbf{x}^*$), i.e.,

$$\mathbb{E}_{p(\theta|\mathbb{D}\cup\{\mathbf{x}^*\})}[f(\theta)] \overset{(9)}{\simeq} \mathbb{E}_{p(\theta|\mathbb{D})}[Mw_\theta f(\theta)] \simeq \tfrac{1}{M}\sum_{\theta\in\Theta} Mw_\theta f(\theta) = \sum_{\theta\in\Theta} w_\theta f(\theta) \qquad (10)$$

for some function $f : \Theta \to \mathbb{R}$. To measure the change in distribution from $p(\theta|\mathbb{D})$ to $p(\theta|\mathbb{D} \cup \{\mathbf{x}^*\})$, we can then use the weights $[w_\theta]_{\theta\in\Theta}$ to compute the *effective sample size* (ESS)

$$\mathrm{ESS}_\Theta(\mathbf{x}^*) = \frac{1}{\sum_{\theta\in\Theta} w_\theta^2} \overset{(9)}{=} \frac{\left(\sum_{\theta\in\Theta} p(\mathbf{x}^*|\theta)\right)^2}{\sum_{\theta\in\Theta} p(\mathbf{x}^*|\theta)^2}, \quad \text{such that} \quad \mathrm{ESS}_\Theta(\mathbf{x}^*) \in [1, M] . \qquad (11)$$

The $\mathrm{ESS}(\mathbf{x}^*)$ is a widely used measure for the efficiency of the estimator in Eq. (10), and measures how many i.i.d. samples drawn from the target posterior $p(\theta|\mathbb{D}\cup\{\mathbf{x}^*\})$ are equivalent to the $M$ samples $\theta \in \Theta$ drawn from the proposal posterior $p(\theta|\mathbb{D})$ and weighted according to $w_\theta$. It intuitively

quantifies the degree of agreement between the particles $\theta \in \Theta$ as to how probable the input $\mathbf{x}^*$ is, inducing the following decision rule: If $\mathrm{ESS}(\mathbf{x}^*)$ is large, then $[w_\theta]_{\theta \in \Theta}$ is close to the uniform distribution $[\frac{1}{M}]_{\theta \in \Theta}$ (for which $\mathrm{ESS}(\mathbf{x}^*) = M$), meaning that all particles $\theta \in \Theta$ explain $\mathbf{x}^*$ equally well and are in agreement as to how probable $\mathbf{x}^*$ is. Thus, $\mathbf{x}^*$ likely is an in-distribution input. Conversely, if $\mathrm{ESS}(\mathbf{x}^*)$ is small, then $[w_\theta]_{\theta \in \Theta}$ contains a few large weights (i.e., corresponding to particles that by chance happen to explain the datum well), with all other weights being very small, where in the extreme case, $[w_\theta]_{\theta \in \Theta} = [0, \ldots, 0, 1, 0, \ldots, 0]$ (for which $\mathrm{ESS}(\mathbf{x}^*) = 1$). This means that the particles do not agree as to how probable $\mathbf{x}^*$ is, so that $\mathbf{x}^*$ likely is an OoD input.

Finally, we discuss how to compute the likelihood $p(\mathbf{x}|\theta)$ of the BVAE given an input $\mathbf{x}$ required to compute the weight $w_\theta$ in Eq. (9). We consider two approaches based on importance sampling, with different proposal distributions for sampling the latent codes $\mathbf{z}_k$ depending on the assumed relationship between the samples $\Theta$ of decoder parameters and the samples $\Phi$ of encoder parameters:

1. Firstly, one may treat the samples in $\Theta$ and $\Phi$ as being coupled as $(\theta_m, \phi_m) \in \{(\theta_m, \phi_m)\}_{m=1}^M$, where each pair $(\theta_m, \phi_m)$ effectively defines a separate VAE. This is motivated by the fact that in practice, we for simplicity do indeed take samples of full VAEs, by simultaneously sampling from $p(\theta|\mathbb{D})$ and $p(\phi|\mathbb{D})$, as outlined in Algorithm 1 in Appendix B. Since such a coupled sample $(\theta_m, \phi_m)$ defines a VAE, the likelihood can then simply be computed as in a regular VAE, i.e., $p(\mathbf{x}|\theta_m) = p(\mathbf{x}|\theta_m, \phi_m)$ as defined in Eq. (2), using the proposal distribution $q(\mathbf{z}|\mathbf{x}, \phi_m)$.

2. Secondly, one may decouple the samples in $\Theta$ and $\Phi$ and treat them as being independent of each other. In particular, for a given decoder sample $\theta_m$, instead of using the corresponding sample $\phi_m$ as the encoder, we may view the encoder as the mixture $q(\mathbf{z}|\mathbf{x}, \mathbb{D}) \simeq \frac{1}{M} \sum_{\phi \in \Phi} q(\mathbf{z}_k|\mathbf{x}, \phi)$ in Eq. (6) defined by all samples in $\Phi$. The likelihood is then estimated as $p(\mathbf{x}|\theta) = p(\mathbf{x}|\theta, \mathbb{D})$ with

$$ p(\mathbf{x}|\theta, \mathbb{D}) = \mathbb{E}_{q(\mathbf{z}|\mathbf{x}, \mathbb{D})} \left[ \frac{p(\mathbf{x}|\mathbf{z}, \theta) p(\mathbf{z})}{q(\mathbf{z}|\mathbf{x}, \mathbb{D})} \right] \overset{(6)}{\simeq} \frac{1}{K} \sum_{k=1}^K \frac{p(\mathbf{x}|\mathbf{z}_k, \theta) p(\mathbf{z}_k)}{\frac{1}{M} \sum_{\phi \in \Phi} q(\mathbf{z}_k|\mathbf{x}, \phi)}, \quad \mathbf{z}_k \sim q(\mathbf{z}|\mathbf{x}, \mathbb{D}) \quad (12) $$

where the proposal distribution now is the mixture $q(\mathbf{z}|\mathbf{x}, \mathbb{D})$ instead of $q(\mathbf{z}|\mathbf{x}, \phi)$. Since $q(\mathbf{z}|\mathbf{x}, \mathbb{D})$ marginalizes over encoder parameters $\phi$, the likelihood in Eq. (12) only depends on $\theta$.

### 3.4 Detecting Outliers in Latent Space

Detecting outliers in latent space involves identifying latent vectors $\mathbf{z}^*$ which are very different from the latent vectors corresponding to the training inputs $\mathbb{D}$, and which will thus yield unpredictable results when passed through the decoder to produce $p(\mathbf{x}|\mathbf{z}^*, \theta)$. To discriminate if a given latent vector $\mathbf{z}^*$ is an outlier or not, we follow the same approach as in Section 3.3 and quantify the informativeness of $\mathbf{z}^*$ on the model parameters $\theta$ by measuring the change in the posterior $p(\theta|\mathbb{D})$. However, quantifying this change in posterior by e.g. computing the ESS metric in Eq. (11) requires evaluating the likelihood $p(\mathbf{x}^*|\theta)$ of the input $\mathbf{x}^*$ corresponding to the latent code $\mathbf{z}^*$, which is not available to us. We thus alternatively quantify the *expected* change in the posterior $p(\theta|\mathbb{D})$ by computing the *expected* ESS, by averaging over the likelihood mixture $p(\mathbf{x}|\mathbf{z}^*, \mathbb{D})$ (cf. Section 3.2), thus effectively taking into account all possible inputs $\mathbf{x}^*$ corresponding to $\mathbf{z}^*$, i.e.,

$$ \mathrm{ESS}_\Theta(\mathbf{z}^*) = \mathbb{E}_{p(\mathbf{x}|\mathbf{z}^*, \mathbb{D})} \left[ \mathrm{ESS}_\Theta(\mathbf{x}) \right] \overset{(11)}{\simeq} \frac{1}{N} \sum_{n=1}^N \frac{\left( \sum_{\theta \in \Theta} p(\mathbf{x}_n|\mathbf{z}^*, \theta) \right)^2}{\sum_{\theta \in \Theta} p(\mathbf{x}_n|\mathbf{z}^*, \theta)^2}, \quad \mathbf{x}_n \sim p(\mathbf{x}|\mathbf{z}^*, \mathbb{D}) \quad (13) $$

where the inputs $\mathbf{x}_n$ are sampled from the decoder mixture $p(\mathbf{x}|\mathbf{z}^*, \mathbb{D})$ by first sampling the index of the mixture component uniformly at random, i.e., $m \sim \mathcal{U}[1, M]$ (due to the mixture weights all being $1/M$), and then sampling $\mathbf{x}_n$ from the $m$-th mixture component, i.e., $\mathbf{x}_n \sim p(\mathbf{x}|\mathbf{z}^*, \theta_m)$. In contrast to computing the ESS in Eq. (11) for outlier detection in input space, where we had to estimate the likelihoods via importance sampling using the (mixture of) inference network(s), computing the likelihoods $p(\mathbf{x}_n|\mathbf{z}^*, \theta)$ in Eq. (13) simply involves passing the given latent code $\mathbf{z}^*$ through the decoders parameterized by $\theta \in \Theta$. In fact, the encoding part of the model is not required for outlier detection in latent space. If we are only interested in outlier detection in latent space, we thus do not face any of the issues arising from the recognition network making wrong inferences for OoD inputs, and as a result do not require a Bayesian treatment of the encoder parameters $\phi$.

## 4 RELATED WORK

**Supervised/Discriminative outlier detection methods**   Most existing OoD detection approaches are task-specific and designed for the context of a certain prediction task. As described in Section 1, these approaches train a deep discriminative model in a supervised fashion using the given labels. To detect outliers w.r.t. the target task, many approaches then rely on a confidence score to decide on the reliability of the prediction, which is either produced by modifying the model and/or training procedure, or computed/extracted post-hoc from the model and/or predictions (DeVries & Taylor, 2018; An & Cho, 2015; Sölch et al., 2016; Ahmed & Courville, 2019; Sricharan & Srivastava, 2018; Hendrycks & Gimpel, 2016; Hendrycks et al., 2018; Liang et al., 2017; Shafaei et al., 2018). Alternatively, one can use predictive uncertainty estimates for OoD detection (Malinin & Gales, 2018; Gal & Ghahramani, 2016; Lakshminarayanan et al., 2017; Osawa et al., 2019; Ovadia et al., 2019). One drawback of such task-specific approaches is that discriminatively trained models by design discard all input features which are not informative about the specific prediction task, such that information which is relevant for OoD detection might be lost.

**Unsupervised/Generative outlier detection methods**   In contrast, task-agnostic OoD detection methods solely use the input data for the unsupervised training of a deep generative model, which makes them more general, as they do not require the availability of labels for some prediction task. Moreover, these approaches capture the entire input data distribution via the generative model, which might be beneficial for OoD detection (it was shown that despite not exploiting label information, unsupervised approaches often work better in practice (Ren et al., 2019)). Perhaps closest to our approach is the recent work by Choi & Jang (2018), which use an ensemble (Lakshminarayanan et al., 2017) of independently trained likelihood-based generative models (i.e., with random parameter initializations and random data shuffling) to approximate the Watanabe-Akaike Information Criterion (Watanabe, 2010) $\mathbb{E}_{p(\theta|\mathbb{D})}[\log p(\mathbf{x}^*|\theta)] - \mathrm{Var}_{p(\theta|\mathbb{D})}[\log p(\mathbf{x}^*|\theta)]$. The WAIC provides an asymptotically correct estimate between the training and test set expectations (assuming a fixed underlying data distribution) and can be viewed as a corrected log-likelihood score which penalizes points $\mathbf{x}$ with a large variance in log-likelihoods. Their approach has two shortcomings as compared to ours: Firstly, while we use stochastic gradient MCMC on a single model to obtain $M$ approximate posterior samples, they train $M$ independent models to form an ensemble, which (a) is not a principled way of doing approximate Bayesian inference (although connections can be made (Mandt et al., 2017) under certain conditions), and (b) is $M$ times more computationally expensive in practice. Secondly, independent of the quality of the posterior approximation, their score is not expected to work well, since it still relies on the suboptimal log-likelihood estimates (as we will demonstrate in the experiments). Another recent unsupervised approach is the likelihood ratio method by Ren et al. (2019), which corrects the likelihood $\log p(\mathbf{x}^*|\theta)$ for confounding general population level background statistics captured by a background model $p(\mathbf{x}^*|\theta_0)$, resulting in the score $\log p(\mathbf{x}^*|\theta) - \log p(\mathbf{x}^*|\theta_0)$. The background model $p(\mathbf{x}^*|\theta_0)$ is in practice trained by perturbing the data $\mathbb{D}$ with noise to corrupt its semantic structure, i.e., by sampling input dimensions i.i.d. from a Bernoulli distribution with rate $\mu \in [0.1, 0.2]$ and replacing their values by uniform noise, e.g. $x_i \sim \mathcal{U}\{0, \ldots, 255\}$ for images. Finally, Nalisnick et al. (2019) proposed to use the score $\left|\log p(\mathbf{x}^*|\theta) - \frac{1}{N}\sum_{\mathbf{x}\in\mathbb{D}} \log p(\mathbf{x}|\theta)\right|$ to account for the typicality of the input $\mathbf{x}^*$.

## 5 EXPERIMENTS

We now present some experimental results showing the competitiveness of our approach.

### 5.1 OUT-OF-DISTRIBUTION DETECTION IN INPUT SPACE

**BVAE details.**   We implemented the proposed BVAE model in `PyTorch` (see Appendix B for pseudocode), using the robust, scale adapted SGHMC variant proposed by Springenberg et al. (2016)[2]. We follow Chen et al. (2014); Springenberg et al. (2016) and use Gaussian priors over $\theta$ and $\phi$, i.e., $p(\theta) = \mathcal{N}(0, \lambda_\theta^{-1})$ and $p(\phi) = \mathcal{N}(0, \lambda_\phi^{-1})$. We also place Gamma hyperpriors over the precision parameters $\lambda_\theta$ and $\lambda_\phi$, i.e., $p(\lambda_\theta) = \Gamma(\alpha_\theta, \beta_\theta)$ and $p(\lambda_\phi) = \Gamma(\alpha_\phi, \beta_\phi)$, with

---

[2]We use their implementation of the SGHMC sampler as a `PyTorch Optimizer` (`https://github.com/automl/pybnn`), thus serving as a drop-in replacement for an optimizer such as SGD.

$\alpha_\theta = \beta_\theta = \alpha_\phi = \beta_\phi = 1$, and resample $\lambda_\theta$ and $\lambda_\phi$ after every training epoch (i.e., after an entire pass over $\mathbb{D}$). We also follow the previous work and use a step size of $10^{-3}$ and momentum decay of $0.05$ for SGHMC. We discard samples within a burn-in phase of $B = 1$ epoch, and store a sample (of both encoder and decoder parameters) after every $D = 1$ epoch. We only keep the 10 most recent samples to represent the posterior. In addition to the BVAE described in Section 3.2 which uses the two ELBOs $\mathcal{L}_{\theta,\phi}$ in Eq. (1) and $\mathcal{L}_\theta$ in Eq. (7) to train/sample from $\phi$ and $\theta$, respectively (denoted by BVAE), we also report results on the following variants: a BVAE that uses the ordinary VAE ELBO in Eq. (1) for both $\phi$ and $\theta$ ($\widetilde{\text{BVAE}}$; see Appendix C for pseudocode) and a VAE where only $\theta$ is integrated out and $\phi$ is estimated via maximum likelihood (BVAE$_\theta$). We furthermore consider both alternative ways of estimating the log-likelihoods for computing the ESS (cf. Section 3.3), i.e., the case where log-likelihoods are computed using samples $\Phi$ of encoders and $\Theta$ of decoders that are treated as independent (ESS), and the case where the log-likelihoods are computed using coupled samples $\{(\phi_j, \theta_j)\}_{j=1}^M$ (ESS$_\phi^\theta$).

**Experimental setup.** We use two benchmarks: (a) FashionMNIST (in-distribution) vs. MNIST (OoD) (Hendrycks et al., 2018; Nalisnick et al., 2018; Ren et al., 2019; Zenati et al., 2018; Akcay et al., 2018), and (b) eight classes of FashionMNIST (in-distribution) vs. the remaining two classes (OoD), using five different splits $\{(0,1), (2,3), (4,5), (6,7), (8,9)\}$ of held-out classes (Ahmed & Courville, 2019). We compare against the log-likelihood (LL) as well as three state-of-the-art methods for unsupervised OoD detection (all described in Section 4): (1) The generative ensemble based method by Choi & Jang (2018) composed of five independently trained models (WAIC), (2) the likelihood ratio method by Ren et al. (2019) (LLR), using Bernoulli rates $\mu = 0.2$ for the Fashion-MNIST vs. MNIST experiment (Ren et al., 2019), and $\mu = 0.15$ for the other experiment, and (3) the test for typicality by Nalisnick et al. (2019) (TT). All methods use VAEs for the log-likelihood estimation.[3] They are evaluated by randomly selecting 5000 in-distribution and OoD inputs from held-out test sets and computing the following threshold independent metrics (Hendrycks & Gimpel, 2016; Liang et al., 2017; Hendrycks et al., 2018; Alemi et al., 2018; Ren et al., 2019): (i) The area under the ROC curve (AUROC↑), (ii) the area under the precision-recall curve (AUPRC↑), and (iii) the false-positive rate at $80\%$ true-positive rate (FPR80↓).

**Results.** Table 1 shows that our approaches generally perform better on the considered benchmarks. Coupling the encoder and decoder samples appears beneficial, which might be due to the fact that we actually take the samples in couples after every epoch. Being Bayesian about the encoder parameters $\phi$ also seems to help. Fig. 2 show the precision-recall and ROC curves used to compute the metrics, for the FashionMNIST vs. MNIST experiment (see Appendix E for the curves for the other experiment). Fig. 1 shows histograms of the log-likelihoods (left) and of the BVAE-ESS scores (right) on FashionMNIST in-distribution (blue) vs. MNIST OoD (orange) test data. While the log-likelihoods strongly overlap, the ESS scores more clearly separate in-distribution data (closer to ESS = 10 on the r.h.s.) from OoD data (closer to ESS = 1 on the l.h.s.).

## 5.2 OUT-OF-DISTRIBUTION DETECTION IN LATENT SPACE

While input space OoD detection is a well-studied problem, OoD detection in latent space was only recently identified as an important open problem (Gómez-Bombarelli et al., 2018; Griffiths & Hernández-Lobato, 2017; Mahmood & Hernández-Lobato, 2019; Alperstein et al., 2019) (see also Section 1). Thus, there is a lack of suitable experimental benchmarks, making a quantitative evaluation challenging. A major issue in designing toy benchmarks based on commonly-used datasets such as MNIST or FashionMNIST (i.e., as typically used for assessing input space OoD detection) is that it is not clear how to obtain ground truth labels for which latent points are OoD and which are not, as we require OoD labels for all possible latent vectors $\mathbf{z}^* \in \mathbb{R}^d$, not just for those corresponding to inputs $\mathbf{x}^*$ from some training or test set. As a first step to facilitate a systematic empirical evaluation of latent space OoD detection techniques, we propose the following experimental protocol.

We use the BVAE variant where only $\theta$ is integrated out and $\phi$ is estimated via maximum likelihood (denoted by BVAE$_\theta$ in Section 5.1, using the same prior and hyperprior as described there). We

---

[3] The architecture of the deconvolutional VAE we use and the training protocol follows previous work (Nalisnick et al., 2018; Choi & Jang, 2018; Ren et al., 2019). We use the Adam optimizer (Kingma & Ba, 2014) with learning rate $10^{-3}$ for all maximum likelihood fits of parameters.

Table 1: AUROC↑, AUPRC↑, and FPR80↓ scores (where higher ↑ or lower ↓ is better) of the baselines (top) and our methods (bottom). For the experiment on FashionMNIST with with held-out classes, we report the mean scores over all five class splits.

| | FashionMNIST vs MNIST | | | FashionMNIST (held-out) | | |
|---|---|---|---|---|---|---|
| | AUROC↑ | AUPRC↑ | FPR80↓ | AUROC↑ | AUPRC↑ | FPR80↓ |
| LL | 0.557 | 0.564 | 0.703 | 0.565 | 0.577 | 0.683 |
| WAIC | 0.541 | 0.548 | 0.798 | 0.446 | 0.464 | 0.827 |
| LLR | 0.617 | 0.613 | 0.638 | 0.560 | 0.569 | 0.698 |
| TT | 0.482 | 0.502 | 0.833 | 0.482 | 0.496 | 0.806 |
| BVAE-ESS | 0.842 | 0.830 | 0.245 | 0.660 | 0.650 | 0.577 |
| BVAE-ESS$_\phi^\theta$ | 0.871 | 0.855 | 0.190 | **0.697** | **0.681** | **0.534** |
| $\widetilde{\text{BVAE-ESS}}$ | 0.897 | 0.893 | 0.149 | 0.672 | 0.659 | 0.562 |
| $\widetilde{\text{BVAE-ESS}}_\phi^\theta$ | **0.921** | **0.907** | **0.082** | 0.683 | 0.668 | 0.558 |
| BVAE$_\theta$-ESS | 0.904 | 0.891 | 0.117 | 0.693 | 0.680 | 0.540 |

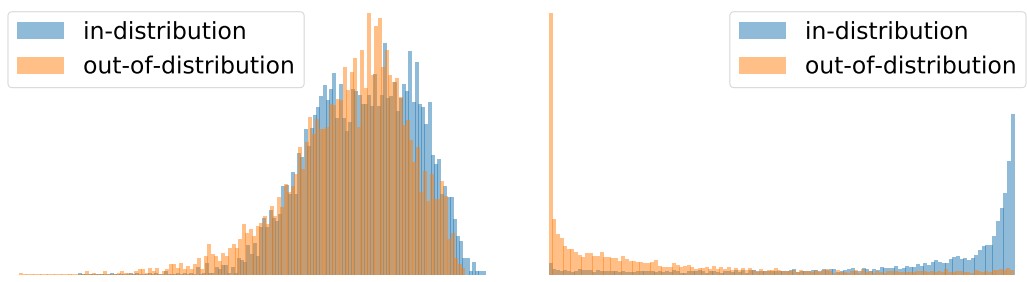

Figure 1: Histograms of LL (left) and $\widetilde{\text{BVAE-ESS}}_\phi^\theta$ (right) on FashionMNIST (in-distribution) and MNIST (out-of-distribution). The ESS score separates the data more clearly than the LL score.

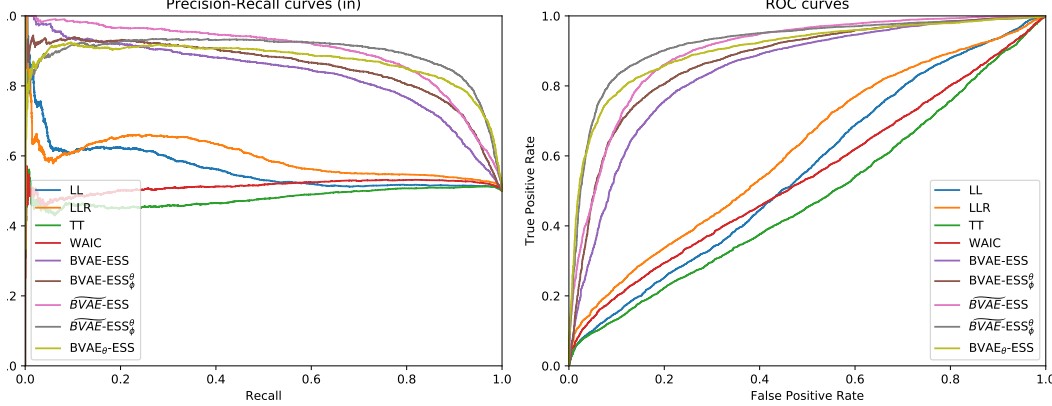

Figure 2: (Left) Precision-recall curves (treating OoD data as the upper class) and (right) ROC curves of all methods on the FashionMNIST vs. MNIST benchmark.

train this BVAE model on some dataset (we will use FashionMNIST here), and then sample $N = 10000$ latent test vectors $\mathbf{z}^*$ from the Gaussian $\mathcal{N}(\mathbf{0}, b \cdot \mathbb{I}_d)$ where $b \in \mathbb{R}^+$, following Mahmood & Hernández-Lobato (2019) (we use $b = 10000$). Since there does not exist a ground truth label for whether a given latent point $\mathbf{z}^*$ is OoD or not, we instead compute some OoD proxy score (to be detailed below) for each of the $N$ latent test vectors and then simply define the $N/2$ latents with the lowest scores to be in-distribution, and the $N/2$ latents with the highest scores to be OoD.

For this OoD score, we propose to train an ensemble of $J$ convolutional neural network classifiers with parameters $\mathbf{W} = \{\mathbf{w}_j\}_{j=1}^J$ on FashionMNIST (Lakshminarayanan et al., 2017). We then approximate the mutual information score[4] proposed by Houlsby et al. (2011), i.e., $\mathbb{I}(\mathbf{w}, y|\mathbf{x}^*, \mathbb{D}) = \mathbb{H}(p(y|\mathbf{x}^*, \mathbb{D})) - \mathbb{E}_{p(\mathbf{w}|\mathbb{D})}[\mathbb{H}(p(y|\mathbf{x}^*, \mathbf{w}))]$, where $\mathbb{E}_{p(\mathbf{w}|\mathbb{D})}[\mathbb{H}(p(y|\mathbf{x}^*, \mathbf{w}))] \simeq \frac{1}{J}\sum_{\mathbf{w}\in\mathbf{W}}\mathbb{H}(p(y|\mathbf{x}^*, \mathbf{w}))$ is the average entropy of the predictive class distribution of the classifier with parameters $\mathbf{w}$, and $\mathbb{H}(p(y|\mathbf{x}^*, \mathbb{D})) \simeq \mathbb{H}\left(\frac{1}{J}\sum_{\mathbf{w}\in\mathbf{W}}p(y|\mathbf{x}^*, \mathbf{w})\right)$ is the entropy of the mixture $\frac{1}{J}\sum_{\mathbf{w}\in\mathbf{W}}p(y|\mathbf{x}^*, \mathbf{w})$ of categorical distributions $p(y|\mathbf{x}^*, \mathbf{w})$ (which is again categorical with averaged probits). Note that this score requires a test input $\mathbf{x}^*$. Since in our setting, we only know the latent code $\mathbf{z}^*$ corresponding to $\mathbf{x}^*$, we instead use the *expected* mutual information under the mixture decoding distribution $p(\mathbf{x}|\mathbf{z}^*, \mathbb{D})$ defined by the decoder ensemble $\Theta$, i.e., $\mathbb{E}_{p(\mathbf{x}|\mathbf{z}^*, \mathbb{D})}[\mathbb{I}(\mathbf{w}, y|\mathbf{x}, \mathbb{D})] \simeq \frac{1}{N}\sum_{n=1}^N \mathbb{I}(\mathbf{w}, y|\mathbf{x}_n, \mathbb{D})$ where $\mathbf{x}_n \sim p(\mathbf{x}|\mathbf{z}^*, \mathbb{D})$. In practice, we use an ensemble of $J = 5$ classifiers and $N = 32$ input samples for the expectation.

We compare the expected ESS (see Section 3.4, with $N = 32$) against two baselines: (a) The distance of $\mathbf{z}^* \in \mathbb{R}^d$ to the spherical annulus of radius $\sqrt{d-1}$, since that is where most probability mass lies under our prior $\mathcal{N}(\mathbf{0}, \mathbb{I}_d)$ (Annulus) (Alperstein et al., 2019). (b) the log-probability of $\mathbf{z}^*$ under the training data distribution in latent space $q(\mathbf{z}) = \frac{1}{N}\sum_{\mathbf{x}\in\mathbb{D}}q(\mathbf{z}|\mathbf{x}, \phi)$, i.e., a uniform mixture of $N$ Gaussians in our case (qz)[5] (Mahmood & Hernández-Lobato, 2019). Fig. 3 shows that our proposed method significantly outperforms the two baselines on this benchmark task.

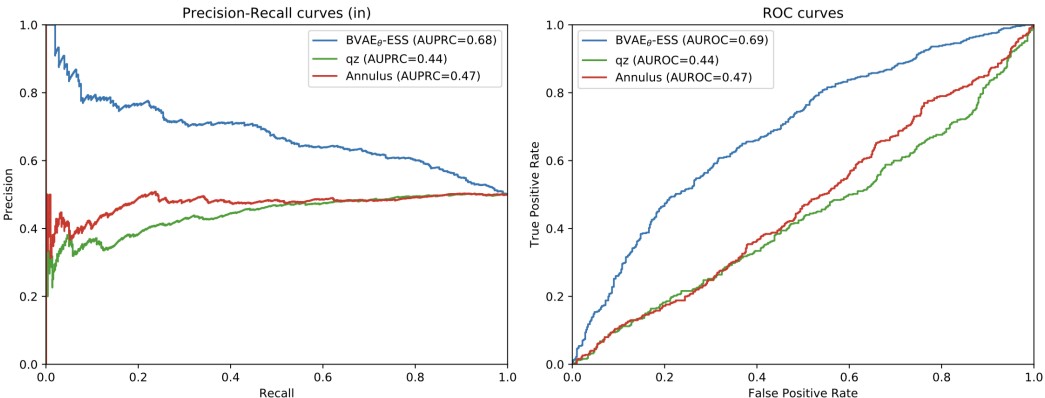

Figure 3: (Left) Precision-recall curves (treating OoD data as the upper class) and (right) ROC curves of all methods on the FashionMNIST based latent space OoD detection benchmark.

## 6 CONCLUSION

We proposed a new approach to unsupervised out-of-distribution detection in input space as well as in latent space, using information-theoretic metrics based on the approximated posterior over the parameters of a variational autoencoder. For future work, we are keen to explore extensions of our approach to (i) state-of-the-art VAE architectures (e.g., using more sophisticated posteriors (Rezende & Mohamed, 2015), priors (Tomczak & Welling, 2017; Papamakarios et al., 2017; Bauer & Mnih, 2019), decoders, etc., which is all complementary); (ii) use alternative lower bounds to the marginal log-likelihood, such as the importance weighted lower bound used in an IWAE (Burda et al., 2015), or combinations of the VAE and IWAE lower bounds, as proposed in (Rainforth et al., 2018); (iii) alternative approximate inference techniques for estimating the posterior distribution over model parameters; (iv) other likelihood-based deep generative models (e.g., flow-based and auto-regressive deep generative models); (v) exploit label information (e.g., by using a hybrid model combining our BVAE with a predictive model, to enable semi-supervised OoD detection).

---

[4]This score is closely related to the disagreement score $\sum_{\mathbf{w}\in\mathbf{W}}D_{\mathrm{KL}}(p(y|\mathbf{x}^*, \mathbf{w})\|p(y|\mathbf{x}^*, \mathbb{D}))$ proposed by Lakshminarayanan et al. (2017) for ensemble-based OoD detection, which could be used alternatively.

[5]For efficiency, we only consider the 100 nearest neighbors (found by a 100-NN model) of a latent test point $\mathbf{z}^*$ for computing this log-probability (Mahmood & Hernández-Lobato, 2019).

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

## A    SVHN VS. CIFAR10 RESULTS FOR OOD DETECTION IN INPUT SPACE

We performed additional experiments on OoD detection in input space on the higher-dimensional SVHN (in-distribution) vs. CIFAR10 (OoD) benchmark[6] (Hendrycks et al., 2018; Nalisnick et al., 2019; Choi & Jang, 2018), following the same experimental protocol as described in Section 5.1. Looking at Table 2 and Fig. 5, we arrive at the same conclusion as with the benchmarks considered in Section 5.1 in that our approaches appear to significantly outperform the baselines we compare against. While the variants BVAE-ESS and BVAE-ESS$_\phi^\theta$ are not as strong as the other BVAE-ESS variants on this benchmark (as compared to the benchmarks considered in Section 3.3), they still outperform the other baselines. The histograms in Fig. 4 again show that the BVAE-ESS scores (right) more clearly separate in-distribution data from OoD data than the log-likelihood scores (left).

Table 2: AUROC↑, AUPRC↑, and FPR80↓ scores (where higher ↑ or lower ↓ is better) of the baselines (top) and our methods (bottom) on the SVHN vs. CIFAR10 benchmark.

|  | AUROC↑ | AUPRC↑ | FPR80↓ |
|---|---|---|---|
| LL | 0.574 | 0.575 | 0.634 |
| WAIC | 0.293 | 0.380 | 0.912 |
| LLR | 0.570 | 0.570 | 0.638 |
| TT | 0.395 | 0.428 | 0.859 |
| BVAE-ESS | 0.667 | 0.656 | 0.564 |
| BVAE-ESS$_\phi^\theta$ | 0.669 | 0.647 | 0.562 |
| $\widetilde{\text{BVAE-ESS}}$ | **0.828** | **0.817** | **0.281** |
| $\widetilde{\text{BVAE-ESS}}_\phi^\theta$ | 0.814 | 0.799 | 0.310 |
| BVAE$_\theta$-ESS | 0.807 | 0.793 | 0.331 |

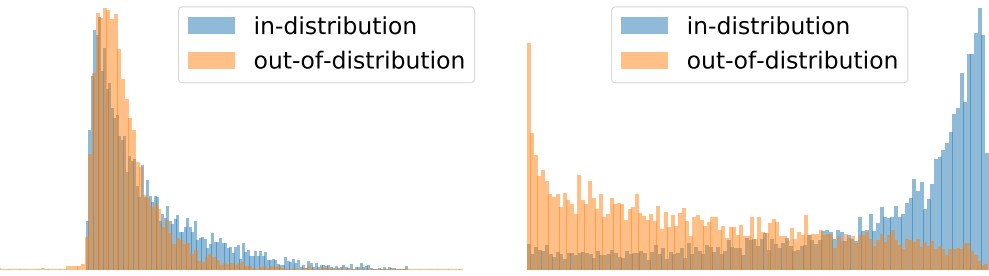

Figure 4: Histograms of LL (left) and $\widetilde{\text{BVAE-ESS}}$ (right) on SVHN (in-distribution) and CIFAR10 (out-of-distribution). The ESS score separates the data more clearly than the LL score.

## B    PSEUDOCODE OF BVAE TRAINING PROCEDURE

Algorithm 1 shows the overall Bayesian VAE (BVAE) training procedure, which produces sets $\Phi = \{\phi_B, \phi_{B+D}, \phi_{B+2D}, \ldots, \phi_T\}$ and $\Theta = \{\theta_B, \theta_{B+D}, \theta_{B+2D}, \ldots, \theta_T\}$ of $M = (T - B)/D + 1$ sequentially generated posterior samples $\phi_t \sim p(\phi|\mathbb{D})$ and $\theta_t \sim p(\theta|\mathbb{D})$ that do not fall into the Markov chain's burn-in phase of $B$ epochs and are $D$ epochs apart from each other (to avoid correlation). Note that Algorithm 1 can in practice be conveniently implemented by exploiting automatic differentiation tools commonly employed by modern deep learning frameworks.

---

[6]Interestingly, following the implementation details provided in Nalisnick et al. (2019), we were not quite able to reproduce their results. In particular, our experiments yielded better calibrated log-likelihood scores for the benchmark CIFAR10 (in-distribution) vs. SVHN (OoD) than theirs (i.e., our VAE model assigned lower likelihood to most of the SVHN OoD data than to the CIFAR10 in-distribution data, as desired); this phenomenon requires further investigation. For this reason, we instead chose the opposite benchmark, i.e., SVHN (in-distribution) vs. CIFAR10 (OoD), for which the likelihood scores largely overlap, as shown in Fig. 4 (left).

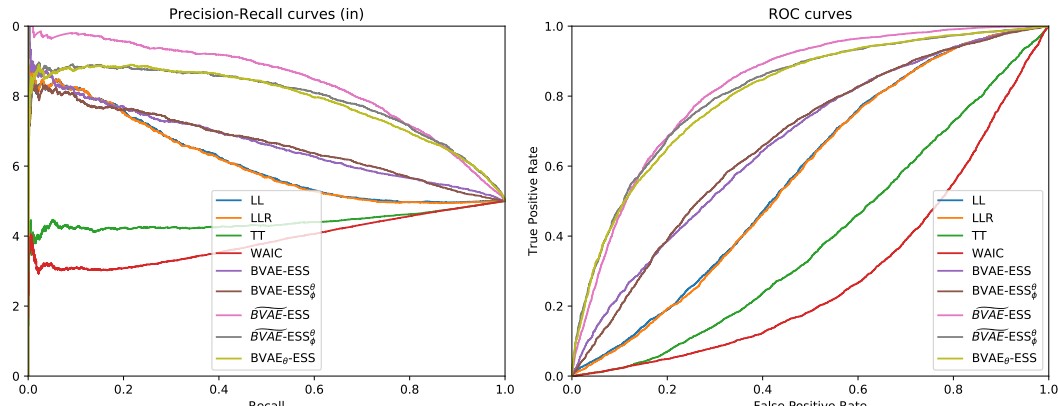

Figure 5: (Left) Precision-recall curves (treating OoD data as the upper class) and (right) ROC curves of all methods on the SVHN vs. CIFAR10 benchmark.

---

**Algorithm 1** Bayesian VAE Training

---

**Input:** Dataset $\mathbb{D}$, generative model $p(\mathbf{x}, \mathbf{z}, \theta)$, inference model $q(\mathbf{z}|\mathbf{x}, \phi)$, burn-in length $B$, sample distance $D$, mini-batch size $|\mathbb{M}|$
Initialize $\phi_0$, $\theta_0$ and $\Phi = \emptyset$, $\Theta = \emptyset$
Define number of mini-batches per epoch $N_b = \frac{|\mathbb{D}|}{|\mathbb{M}|}$
**for** $t = 1, \ldots, T$ **do**
    Set $\hat{\phi}_0 = \phi_{t-1}, \hat{\theta}_0 = \theta_{t-1}$
    **for** $b = 1, \ldots, N_b$ **do**
        Sample minibatch $\mathbb{M} \sim \mathbb{D}$
        Set $\Theta_* = \{\hat{\theta}_{b-1}\}$ **if** $t \leq B$ **else** $\Theta_* = \Theta$
        Set $\Phi_* = \{\hat{\phi}_{b-1}\}$ **if** $t \leq B$ **else** $\Phi_* = \Phi$
        Compute $\nabla_\phi \tilde{U}(\hat{\phi}_{b-1}, \Theta_*, \mathbb{M})$, update $\hat{\phi}_{b-1} \to \hat{\phi}_b$ via SGHMC
        Compute $\nabla_\theta \tilde{U}(\hat{\theta}_{b-1}, \Phi_*, \mathbb{M})$, update $\hat{\theta}_{b-1} \to \hat{\theta}_b$ via SGHMC
    **end for**
    Set $\phi_t = \hat{\phi}_{N_b}, \theta_t = \hat{\theta}_{N_b}$
    **if** $t \geq B$ and $(t - B) \bmod D = 0$ **then**
        Add $\Phi = \Phi \cup \{\phi_t\}$ and $\Theta = \Theta \cup \{\theta_t\}$
    **end if**
**end for**
**Output:** Posterior samples $\Phi$ and $\Theta$

---

$\nabla_\phi \tilde{U}(\hat{\phi}_{b-1}, \Theta_*, \mathbb{M})$ and $\nabla_\theta \tilde{U}(\hat{\theta}_{b-1}, \Phi_*, \mathbb{M})$ are approximations to the stochastic gradient in Eq. (3) required for the SGHMC updates, defined as

$$\nabla_\phi U(\phi, \mathbb{M}) \approx \nabla_\phi \tilde{U}(\hat{\phi}_{b-1}, \Theta_*, \mathbb{M}) = -\frac{|\mathbb{D}|}{|\mathbb{M}|} \sum_{\mathbf{x} \in \mathbb{M}} \nabla_\phi \mathcal{L}_{\hat{\phi}_{b-1}}(\mathbf{x}) - \nabla_\phi \log p(\hat{\phi}_{b-1}) \qquad (14)$$

with ELBO gradient $\nabla_\phi \mathcal{L}_{\hat{\phi}_{b-1}}(\mathbf{x})$ as in Eq. (5), and

$$\nabla_\theta U(\theta, \mathbb{M}) \approx \nabla_\theta \tilde{U}(\hat{\theta}_{b-1}, \Phi_*, \mathbb{M}) = -\frac{|\mathbb{D}|}{|\mathbb{M}|} \sum_{\mathbf{x} \in \mathbb{M}} \nabla_\theta \mathcal{L}_{\hat{theta}_{b-1}}(\mathbf{x}) - \nabla_\theta \log p(\hat{\theta}_{b-1}) \qquad (15)$$

with ELBO gradient $\nabla_\theta \mathcal{L}_{\hat{\theta}_{b-1}}(\mathbf{x})$ as in Eq. (8).

## C    PSEUDOCODE OF ALTERNATIVE BVAE TRAINING PROCEDURE

Algorithm 2 shows pseudocode of an alternative Bayesian VAE (BVAE) training procedure, which essentially is identical to the training of an ordinary VAE, but using the SGHMC sampler instead of

a stochastic gradient optimizer. The posterior samples $\Theta$ and $\Phi$ are thus not used during training (as in Algorithm 1), but only for test time prediction.

---

**Algorithm 2** Alternative Bayesian VAE Training

---

**Input:** Dataset $\mathbb{D}$, generative model $p(\mathbf{x}, \mathbf{z}, \theta)$, inference model $q(\mathbf{z}|\mathbf{x}, \phi)$, burn-in length $B$, sample distance $D$, mini-batch size $|\mathbb{M}|$
Initialize $\phi_0$, $\theta_0$ and $\Phi = \emptyset$, $\Theta = \emptyset$
Define number of mini-batches per epoch $N_b = \frac{|\mathbb{D}|}{|\mathbb{M}|}$
**for** $t = 1, \ldots, T$ **do**
    Set $\hat{\phi}_0 = \phi_{t-1}$, $\hat{\theta}_0 = \theta_{t-1}$
    **for** $b = 1, \ldots, N_b$ **do**
        Sample minibatch $\mathbb{M} \sim \mathbb{D}$
        Compute $\nabla_\phi \tilde{U}(\hat{\phi}_{b-1}, \hat{\theta}_{b-1}, \mathbb{M})$, update $\hat{\phi}_{b-1} \to \hat{\phi}_b$ via SGHMC
        Compute $\nabla_\theta \tilde{U}(\hat{\phi}_{b-1}, \hat{\theta}_{b-1}, \mathbb{M})$, update $\hat{\theta}_{b-1} \to \hat{\theta}_b$ via SGHMC
    **end for**
    Set $\phi_t = \hat{\phi}_{N_b}$, $\theta_t = \hat{\theta}_{N_b}$
    **if** $t \geq B$ and $(t - B) \bmod D = 0$ **then**
        Add $\Phi = \Phi \cup \{\phi_t\}$ and $\Theta = \Theta \cup \{\theta_t\}$
    **end if**
**end for**
**Output:** Posterior samples $\Phi$ and $\Theta$

---

We have

$$\tilde{U}(\hat{\phi}_{b-1}, \hat{\theta}_{b-1}, \mathbb{M}) = -\frac{|\mathbb{D}|}{|\mathbb{M}|} \sum_{\mathbf{x} \in \mathbb{M}} \mathcal{L}_{\hat{\theta}_{b-1}, \hat{\phi}_{b-1}}(\mathbf{x}) - \log p(\hat{\phi}_{b-1}) - \log p(\hat{\theta}_{b-1}) \qquad (16)$$

with the standard VAE ELBO as in Eq. (1), i.e.,

$$\mathcal{L}_{\hat{\theta}_{b-1}, \hat{\phi}_{b-1}}(\mathbf{x}) = \mathbb{E}_{q(\mathbf{z}|\mathbf{x}, \hat{\phi}_{b-1})}[\log p(\mathbf{x}|\mathbf{z}, \hat{\theta}_{b-1})] - D_{\mathrm{KL}}(q(\mathbf{z}|\mathbf{x}, \hat{\phi}_{b-1})\|p(\mathbf{z})) \ . \qquad (17)$$

## D NUMERICALLY STABLE IMPLEMENTATION OF THE ESS SCORE

For numerical stability, we in practice always work with log-probabilities (since raw probabilities may get arbitrarily close to and thus be rounded to zero). I.e., instead of directly computing $p(\mathbf{x}|\theta)$ in Eq. (12) based on the probabilities $p(\mathbf{x}|\mathbf{z}, \theta)$, $p(\mathbf{z})$ and $q(\mathbf{z}|\mathbf{x}, \phi)$, we compute $\log p(\mathbf{x}|\theta)$ based on the respective log-probabilities $\log p(\mathbf{x}|\mathbf{z}, \theta)$, $\log p(\mathbf{z})$ and $\log q(\mathbf{z}|\mathbf{x}, \phi)$ as follows:

$$\log p(\mathbf{x}|\theta) \simeq \log \left( \frac{1}{K} \sum_{k=1}^{K} \frac{p(\mathbf{x}|\mathbf{z}_k, \theta) p(\mathbf{z}_k)}{\frac{1}{M} \sum_{\phi \in \Phi} q(\mathbf{z}_k|\mathbf{x}, \phi)} \right)$$

$$= \log \left( \frac{1}{K} \sum_{k=1}^{K} \exp \left( \log \frac{p(\mathbf{x}|\mathbf{z}_k, \theta) p(\mathbf{z}_k)}{\frac{1}{M} \sum_{\phi \in \Phi} q(\mathbf{z}_k|\mathbf{x}, \phi)} \right) \right)$$

$$= \mathrm{logmeanexp}_{\mathbf{z}_k} \left( \log \frac{p(\mathbf{x}|\mathbf{z}_k, \theta) p(\mathbf{z}_k)}{\frac{1}{M} \sum_{\phi \in \Phi} q(\mathbf{z}_k|\mathbf{x}, \phi)} \right)$$

$$= \mathrm{logmeanexp}_{\mathbf{z}_k} \left( \log \frac{\exp(\log(p(\mathbf{x}|\mathbf{z}_k, \theta) p(\mathbf{z}_k)))}{\exp(\log(\frac{1}{M} \sum_{\phi \in \Phi} \exp(\log p(\mathbf{z}_k|\mathbf{x}, \phi))))} \right)$$

$$= \mathrm{logmeanexp}_{\mathbf{z}_k} \left( \log \frac{\exp(\log p(\mathbf{x}|\mathbf{z}_k, \theta) + \log p(\mathbf{z}_k))}{\exp(\mathrm{logmeanexp}_\phi(\log p(\mathbf{z}_k|\mathbf{x}, \phi)))} \right)$$

$$= \mathrm{logmeanexp}_{\mathbf{z}_k} \left( \log p(\mathbf{x}|\mathbf{z}_k, \theta) + \log p(\mathbf{z}_k) - \mathrm{logmeanexp}_\phi(\log p(\mathbf{z}_k|\mathbf{x}, \phi)) \right)$$

where $\mathbf{z}_k \sim q(\mathbf{z}|\mathbf{x}, \mathbb{D})$, where the last equality uses $\log \frac{\exp(a)}{\exp(b)} = \log(\exp(a - b)) = a - b$, and where logmeanexp is a variant of the commonly used numerically stable logsumexp function which computes the mean instead of the sum.

We thus obtain the vector $\boldsymbol{p} = [p_\theta]_{\theta \in \Theta} = [\log p(\mathbf{x}|\theta)]_{\theta \in \Theta}$ of log-likelihoods, which we can then use to compute the vector $\boldsymbol{w} = [w_\theta]_{\theta \in \Theta}$ of weights $w_\theta$ in Eq. (9) by passing $\boldsymbol{p}$ through the numerically stable softmax function, i.e.,

$$w_\theta = \frac{p(\mathbf{x}|\theta)}{\sum_{\theta \in \Theta} p(\mathbf{x}|\theta)} = \frac{\exp(p_\theta)}{\sum_{\theta \in \Theta} \exp(p_\theta)} = \frac{\exp(p_\theta - p^*)}{\sum_{\theta \in \Theta} \exp(p_\theta - p^*)} = \mathrm{softmax}(\boldsymbol{p} - p^*)_\theta \quad (18)$$

where $p^* = \max(\boldsymbol{p}) = \max\{\log p(\mathbf{x}|\theta) | \theta \in \Theta\}$ is the largest log-likelihood value across all $\theta \in \Theta$.

# E  PRECISION-ROC AND PR CURVES FOR FASHIONMNIST (HELD-OUT CLASSES) BENCHMARK

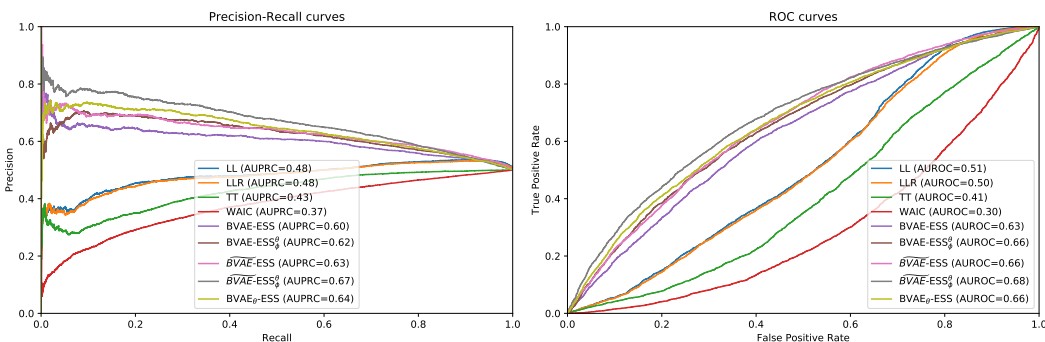

Figure 6: (Left) Precision-recall curves and (right) ROC curves of all methods on the FashionMNIST (held-out classes) benchmark with classes 0 and 1 held-out.

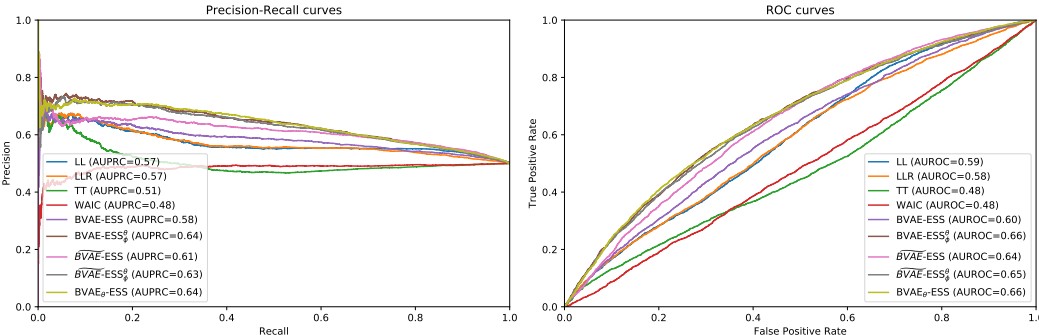

Figure 7: (Left) Precision-recall curves and (right) ROC curves of all methods on the FashionMNIST (held-out classes) benchmark with classes 2 and 3 held-out.

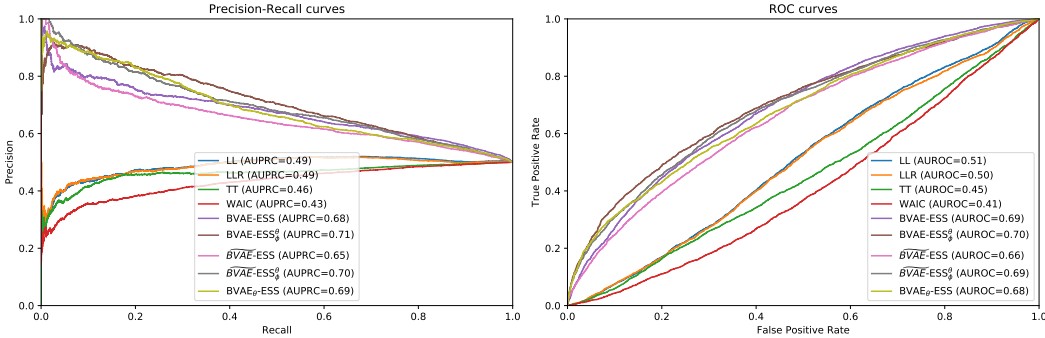

Figure 8: (Left) Precision-recall curves and (right) ROC curves of all methods on the FashionMNIST (held-out classes) benchmark with classes 4 and 5 held-out.

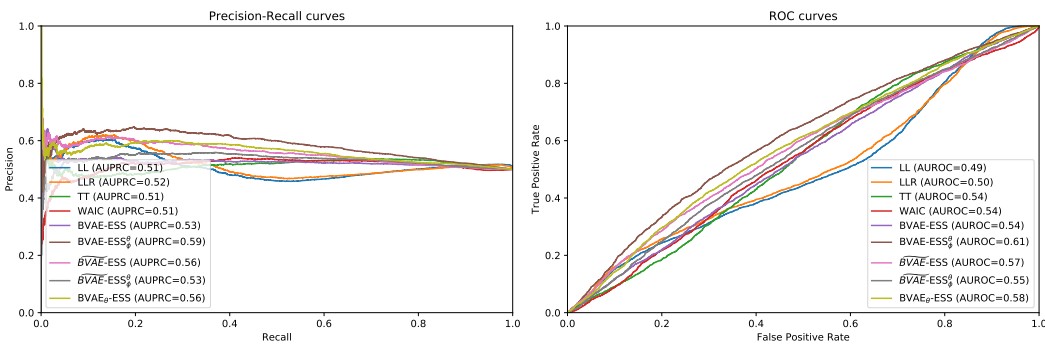

Figure 9: (Left) Precision-recall curves and (right) ROC curves of all methods on the FashionMNIST (held-out classes) benchmark with classes 6 and 7 held-out.

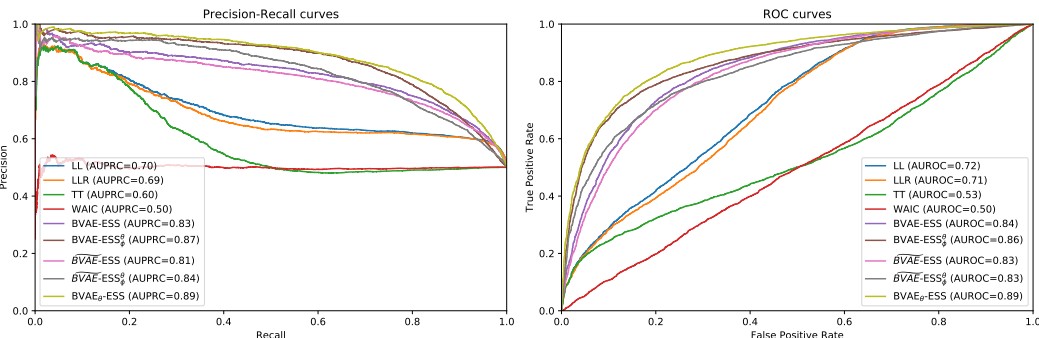

Figure 10: (Left) Precision-recall curves and (right) ROC curves of all methods on the FashionM-NIST (held-out classes) benchmark with classes 8 and 9 held-out.

