# OpenReview forum: "Bayesian Variational Autoencoders for Unsupervised Out-of-Distribution Detection"
_ICLR.cc/2020/Conference — Reject_

### Official Review · AnonReviewer3 · 2019-10-23
**Official Blind Review #3**

**Rating:** 3

**Review:**

The paper advocates to use information gain to detect whether a sample is out of distribution. To that end, a Bayesian VAE is introduced for which that quantity is tractable. The experiments show a solid improvement over previous methods.

I like the paper, but I have a few questions. I am more than willing to increase my score from weak reject to weak or strong accept if these are addressed properly.

What is the relationship of information gain to the marginal likelihood of the data? Since both can be expressed in entropies, I can see a very strong relationship, but would enjoy the authors opinion here–what exactly is it that gives the edge?

The experiments report results on the likelihood based score. Were these results taken from previous publications or obtained from exactly the same pipeline?

Why is the "outlier in latent space" section included even though it is not experimentally verified? I think it should go, as conducting experiments is cheap in ML. On the other hand, if we cannot come up with an experiment to conduct, then what is hypothesis is tested? I think the section needs to be removed and be revisited in future work.

Is the method really principled? Where is the connection from the assumption that the score should be high for out of distribution and low for in distribution? If a method is called "principled" I want to see a rigorous derivation of how a method derives from what principles exactly and how it is approximated.

Since only the 10 most recent samples are kept to represent the posterior, I am worried about their diversity. I think the authors should back up that this is sufficient to represent the posterior.

What happens in the non-parametric limit, where the posterior will collapse to a point? Does the method not rely on an insufficiently inferred model?

**Experience Assessment:**

I have published one or two papers in this area.

**Review Assessment: Checking Correctness Of Derivations And Theory:**

I carefully checked the derivations and theory.

**Review Assessment: Checking Correctness Of Experiments:**

I assessed the sensibility of the experiments.

**Review Assessment: Thoroughness In Paper Reading:**

I read the paper thoroughly.

---

> ### Author Response · Authors · 2019-11-13
> **Author Response**
>
> We thank you for your insightful and constructive feedback, which we will take into account when revising our paper. We address all points you raised below.
>
> 1. "What is the relationship of information gain to the marginal likelihood of the data? Since both can be expressed in entropies, I can see a very strong relationship, but would enjoy the authors opinion here–what exactly is it that gives the edge?"
> While the two quantities are indeed related, they measure two very different things (we will add a discussion of this to the paper -- thanks for pointing this out):
> The marginal likelihood measures the probability that the model gives to the observed data.
> In contrast, the information gain measures the reduction in entropy about the posterior over model parameters after having observed a datum x; viewed differently, the information gain (or the ESS that we use) quantifies the >variation< of the marginal likelihoods of different models under the posterior, which we argue is more indicative for OoD detection than the marginal likelihoods themselves.
>
> 2. "The experiments report results on the likelihood based score. Were these results taken from previous publications or obtained from exactly the same pipeline?"
> For comparability, all results reported in the paper were obtained from our experimental pipeline -- we did not report any results from previous publications.
>
> 3. "Why is the "outlier in latent space" section included even though it is not experimentally verified?"
> We agree that such an evaluation is desirable, even in absence of an established experimental protocol or strong baselines for comparison.
> We thus now designed an experimental protocol and added initial results for out-of-distribution detection in latent space to the paper (see Section 5.2), and plan to add further results (e.g., on more datasets).
> In future work, we plan to apply our method to more complex settings (beyond the scope of this paper), such as molecular design and other applications relying on optimization in latent space of a VAE (as described in the introduction).
>
> 4. "Is the method really principled? Where is the connection from the assumption that the score should be high for out of distribution and low for in distribution? If a method is called "principled" I want to see a rigorous derivation of how a method derives from what principles exactly and how it is approximated."
> We call our method principled for two reasons (which we will clarify in the paper):
> 1) As you correctly mentioned, the ESS score measures the information gain, i.e., the reduction in entropy about the model parameters after having observed an input x.
> Such information-theoretic metrics to quantify the novelty of data points are widely and successfully used in information-theoretic active learning; see e.g. (MacKay, 1992; see our paper for references) or BALD (Houlsby et al, 2011) for motivations for such metrics.
> We argue that the notion of novelty of a datum x captured by the information gain is exactly the notion of novelty required to effectively detect outliers, revealing a fundamental connection between active learning and OoD detection.
> In particular, we argue that such established measures rooted in information theory are more principled than many of the previous OoD detection methods, which are often ad-hoc heuristics.
> 2) We use more principled approximate inference techniques to estimate the posterior over model parameters than previous work such as (Choi et al, 2018), which simply use an ensemble of independently trained models as a proxy for posterior samples.
>
> 5. "Since only the 10 most recent samples are kept to represent the posterior, I am worried about their diversity. I think the authors should back up that this is sufficient to represent the posterior."
> Preliminary experiments (not included in the paper) showed that using more samples and/or a larger thinning interval does not significantly improve performance. We will add a systematic evaluation of this matter to the paper; thank you for the suggestion!
>
> 6. "What happens in the non-parametric limit, where the posterior will collapse to a point? Does the method not rely on an insufficiently inferred model?"
> Are you referring to the case in which the inference network has very large capacity and can approximate the true posterior over latent variables z arbitrarily well?
> In that case, there will indeed not be any (epistemic) uncertainty left in the inference network, such that estimating a posterior over encoder parameters will not help for OoD detection.
> However, there will in general still be parametric uncertainty in the generative network, which can be captured by estimating a posterior distribution over the decoder parameters.
> Even if we can infer the true latent code z corresponding to an input x, the decoder posterior samples will still agree in their likelihood estimates for in-distribution data, and disagree for OoD data, such that our method should still work in this setting.

---

> > ### Comment · AnonReviewer3 · 2019-11-15
> > **reply**
> >
> > Thanks for the reply.
> >
> > IMHO, principled means that a problem is formally posed and the solution is then derived from it. In that sense, information gain is principled for active learning, but it is not for out of distribution–at least I don't see the argument yet.
> >
> > E.g., if we have a single Gaussian, infer its posterior given an infinite data set, then my intuition tells me that the information gain for *all* possible data points will be 0. Hence, the property of OoD detection collapses, because all samples are treated equally. I don't feel convinced.

---

> > > ### Author Response · Authors · 2019-11-15
> > > **Author response**
> > >
> > > Thank you very much for your quick response and for facilitating this discussion!
> > >
> > > We see the point you are trying to make.
> > > The thing is that formalizing the OoD detection problem to allow for a principled solution is somewhat difficult.
> > >
> > > OoD detection is often "formalized" by assuming that we are given a set of samples from some distribution p, and need to decide if a single, previously unseen datum x was sampled from p or from a different distribution q.
> > > Unfortunately, this problem is inherently ill-posed: How "different" are p and q?
> > > Also, can we even draw meaningful conclusions about the distribution of x by only observing a single sample?
> > > Clearly, for arbitrary distributions p and q, there cannot exist a classifier that can perfectly distinguish if x was drawn from p or q (e.g. if their supports overlap).
> > > If we had access to the density of p, then the natural solution would be to classify x based on the probability p(x) (which could be viewed as a principled solution), since OoD data drawn from q should have lower probability under p than the training data drawn from p.
> > >
> > > If we do not have access to the density of p, then there exist the following alternatives:
> > > 1) As you pointed out, we could have access to an infinite number of samples from p, which would allow us to perfectly characterize p (e.g. we could perfectly fit a Gaussian, as you described).
> > > In practice, we will of course never have an infinite amount of data, but let us consider this case for the sake of argument.
> > > You are right that in that case, the information gain would be zero for any point.
> > > However, in that case, we could simply rely on our characterization of p to detect OoD inputs by looking at p(x).
> > > In particular, if the supports of p and q do not overlap, then OoD data drawn from q will have zero probability under p.
> > > In contrast, as soon as we have a lack of information (i.e., only a finite amount of data), then we will not be able to fit a perfect model for p, in which case the information gain will be a useful measure to tell us which data points are likely to be outliers.
> > >
> > > 2) If we do not have infinite data, then we could try to estimate p(x) based on the given samples of p.
> > > However, estimating complex high-dimensional probability distributions from samples is an open problem, and as recent research has shown (Nalisnick et al, 2019, Choi et al, 2018), the deep generative models we typically use fail to provide reliable estimates of p(x), which somewhat invalidates this approach for detecting OoD data.
> > > This motivated some recent work (described in our paper) that tries to correct the likelihood estimate p(x); however, these methods would probably not qualify as being principled under your definition.
> > > Apart from that, there are many other OoD detection methods which do not try to estimate p(x) using a generative model, but instead use other ad-hoc heuristics to tweak a supervised classifier to tackle this problem (which we call supervised/discrmininative OoD detection methods in our paper).
> > > Most (perhaps all) of these methods would probably also not qualify as being principled.
> > > This motivates our work of trying to find an alternative, practical measure to detect OoD inputs which does not rely on the likelihood, but is still as principled as possible.
> > > As there seems to be a discrepancy between how principled a method is vs. how well it works in practice, we believe we found a good trade-off between the two (e.g., our experiments demonstrate while the log-likelihood score might be more principled, it may perform much worse than our approach).
> > >
> > > Also in our view, active learning and OoD detection are very much related, as both problems are concerned with identifying data points which are "different" to all data points we have seen so far (i.e., during training), where this difference can be quantified using information-theoretic measures.
> > > The main difference might be that in active learning, we typically assume that all possible points that we can pick come from the training data distribution, such that the most informative point will help us fit our model; in contrast, in OoD detection, we more generally assume that data might come from a distribution different to the training data distribution, such that the most informative point will likely be OoD. Due to these inherent connections, we argue that principled active learning techniques can also be used for OoD detection
> > >
> > > In conclusion, you are right in that our method might not be principled in your sense. However, as most previous work has focused on devising heuristics, mostly with little/no theoretical justification, we view our work as being at least more principled than previous approaches, by having an information-theoretic justification and strong connections to the very much related active learning problem.
> > > In any case, if you think it might be an overstatement, we are happy to remove the predicate "principled" from the paper!

---

### Official Review · AnonReviewer1 · 2019-10-24
**Official Blind Review #1**

**Rating:** 3

**Review:**

After reading all the reviews, the comments, and the additional work done by the Authors, I have decided to confirm my rating.

==================

This paper leverage probabilistic inference techniques to maintain a posterior distribution over the parameters of a variational autoencoder (VAE). This results in a Bayesian VAE (BVAE) model, where instead of fitting a point estimate of the decoder parameters via maximum likelihood, they estimate their posterior distribution using samples generated via stochastic gradient Markov chain Monte Carlo (MCMC).
The informativeness of an unobserved input x* / latent z* is then quantified by measuring the (expected) change in the posterior over model parameters after having observed x* / z*.  The motivation is clear, when considered inputs which are uninformative about the model parameters, they are likely similar to the data points already in the training set. In contrast, inputs which are very informative about the model parameters are likely different from everything in the training data.

The contributions are:
- A Bayesian VAE model which uses state-of-the-art Bayesian inference techniques to estimate a posterior distribution over the decoder parameters.
- A description of how this model can be used to detect outliers both in input space and in the model’s latent space.
- Results showing that this approach outperforms state-of-the-art outlier detection methods.

The paper is well written, and the proposed ideas are well motivated.
However, the experiment section is too limited. The authors should at least use one more dataset such as CIFAR10. They just use FashionMNIST vs MNIST FashionMNIST (held-out).
In addition, it would strengthen the paper if the authors could show at least initial result about how the model performs to detect out of distribution in the latent space, given that it is considered as part of the contribution.

The paper lacks some references such as:
- Predictive uncertainty estimation via prior networks, NEURIPS 2018.


**Experience Assessment:**

I have published one or two papers in this area.

**Review Assessment: Checking Correctness Of Derivations And Theory:**

I assessed the sensibility of the derivations and theory.

**Review Assessment: Checking Correctness Of Experiments:**

I assessed the sensibility of the experiments.

**Review Assessment: Thoroughness In Paper Reading:**

I read the paper at least twice and used my best judgement in assessing the paper.

---

> ### Author Response · Authors · 2019-11-13
> **Author Response**
>
> Thank you for your helpful comments and suggestions!
>
> 1. "The experiment section is too limited. The authors should at least use one more dataset such as CIFAR10."
> Thank you for this suggestion. We will add additional experimental results on CIFAR10 to the paper (before the end of the rebuttal period on Friday).
>
> 2. "It would strengthen the paper if the authors could show at least initial result about how the model performs to detect out of distribution in the latent space, given that it is considered as part of the contribution".
> Yes, this is a fair point, thank you for the suggestion. We added initial results for out-of-distribution detection in latent space to the paper (see Section 5.2) and plan to add further results (e.g., on more datasets).
>
> 3. "The paper lacks some references such as: Predictive uncertainty estimation via prior networks, NEURIPS 2018."
> Thank you for pointing out this work. We added it to the related work section of our paper, along with a few other works that do out-of-distribution detection by estimating predictive uncertainties.

---

### Official Review · AnonReviewer2 · 2019-10-26
**Official Blind Review #2**

**Rating:** 3

**Review:**

This paper studies the problem of out-of-distribution data detection, which is an important problem in machine learning. The authors propose to use Bayesian variational autoencoder which applies SGHMC to get samples of the weights of the encoder and the decoder. The proposed method is tested on two benchmarks to demonstrate effectiveness.

The proposed Bayesian variational autoencoder appears to be technically sound. When applying it to OoD detection, effective sample size is used to quantify how much the posterior changes given the new data. The authors claim that ESS will be large when the data is in-distribution since all samples explain the data equally well. First, I’m not sure this is true that all samples from the posterior should explain the data equally well even if it is in-distribution. Second, if the data is out of distribution, it is likely that all samples explain the data equally bad which also results in high ESS. In practice, it is very likely that p(x*|theta) are low for all the theta when x* is out-of-distribution. Am I missing something here?

How to determine whether a data is out-of-distribution or not based on ESS? Is the threshold of ESS a hyperparameter to tune?

For the experiments, I wonder why the authors put Gamma hyper priors for BVAE which was not used in the previous work that use SGHMC. Is there any reason for doing this? Again, it is unclear to me how the authors decide whether a data is out-of-distribution or not based on ESS.

It seems like simply applying SGHMC for the decoder parameters is sufficient, as the other treatments only improve the results incrementally but adding large computational and storage cost. I’m not familiar with the literature enough to tell whether the results of previous methods are reasonable or not. By looking at the table, it seems that the proposed method achieves some gain over the previous methods.

In the experiments, BVAE only keeps the most recent 10 samples. Aren’t the samples very similar? Since the thinning interval is only 1 epoch.

It would make the paper stronger if the authors are able to demonstrate the usefulness of detecting OoD in latent space through experiments.


**Experience Assessment:**

I have published one or two papers in this area.

**Review Assessment: Checking Correctness Of Derivations And Theory:**

I assessed the sensibility of the derivations and theory.

**Review Assessment: Checking Correctness Of Experiments:**

I assessed the sensibility of the experiments.

**Review Assessment: Thoroughness In Paper Reading:**

I read the paper at least twice and used my best judgement in assessing the paper.

---

> ### Author Response · Authors · 2019-11-13
> **Author Response**
>
> We want to thank you for your helpful feedback, which will help us to improve our paper. Please see below for our clarifications to the concerns youraised.
>
> 1. "First, I’m not sure this is true that all samples from the posterior should explain the data equally well even if it is in-distribution. Second, if the data is OoD, it is likely that all samples explain the data equally bad which also resultsin high ESS."
> This becomes clearer when looking at the analogous (but more intuitive) supervised regression setting: Consider a Bayesian regression model p(y|x), which captures epistemic uncertainty via a posterior over the model parameters, inducing uncertainty in the predictions y. This model will have >low< predictive uncertainty for an in-distribution input x, meaning that posterior parameter samples explain x equally well and thus >agree< on their predictions y. Conversely, for an OoD input x, the predictive uncertainty (i.e., >variation< in the predictions) will be >high<, i.e., posterior samples >disagree< on their predictions y. Importantly, not all samples will be equally bad at explaining x, but some will (by chance) be better than others.In our setting, we can view the likelihood estimate p(x|theta) as a (implicit) function from inputs x to outputs y = p(x|theta) which is fully specified by the inference and generative networks, such that a Bayesian treatment of the parameters yields a similar effect as in the regression setting. I.e., for an in-distribution input x, the likelihood "predictions" y = p(x|theta) will be similar for different posterior samples (i.e., the samples explain the input equally well). While for an OoD input x, the "predictions" y = p(x|theta) will likely all be bad, the important thing is that their >variation< will be high, i.e., the samples will not explain x equally bad, but some samples will (by chance) explain x better than others (see also our answer to point 2. below).
> Thus, since the ESS measures the variation across likelihoods, we expect it to be a good metric for OoD detection.
> We will add an explanation to the paper.
>
> 2. "In practice, it is very likely that p(x*|theta) are low for all the theta when x* is OoD."
> Unfortunately, it was shown that this is not generally true, as deep generative models might assign higher likelihood to OoD samples than to in-distribution samples (Nalisnick et al, 2019; Choi et al, 2018; see our paper for references).
> As a result, the likelihood cannot be used as a robust measure for OoD detection, which is one of the motivations for our work.
>
> 3. "How to determine whether a data is out-of-distribution or not based on ESS? Is the threshold of ESS a hyperparameter to tune?"
> Yes, in practice, one needs to define a threshold for the ESS score.
> Note, however, that this is not a limitation of our method, as all other scores proposed in the literature also require a threshold.
> There exist some proposals in the literature for defining such a threshold; we will add a discussion of this to the paper.
>
> 4. "For the experiments, I wonder why the authors put Gamma hyper priors for BVAE which was not used in the previous work that use SGHMC. Is there any reason for doing this?"
> In fact, as mentioned in the paper, previous work does propose to use Gamma hyper priors, including the paper introducing SGHMC (Chen et al. 2014; see their Section 4.2 and Appendix H.1).
>
> 5. "Again, it is unclear to me how the authors decide whether a data is out-of-distribution or not based on ESS."
> All metrics we report in our experiments (i.e., AUPRC, AUROC, FPR80) are threshold independent, as they take into account the performance across all possible thresholds.
> This is common practice in the OoD detection literature, to avoid having to specify thresholds for each method.
>
> 6. "It seems like simply applying SGHMC for the decoder parameters is sufficient, as the other treatments only improve the results incrementally but adding large computational and storage cost."
> While using SGHMC over only the decoder parameters might be sufficient, using SGHMC also over the encoder parameters induces only neglible additional computational cost (SG-MCMC methods are as expensive as stochastic optimizers such as SGD), and only double the memory cost.
>
> 7. "In the experiments, BVAE only keeps the most recent 10 samples. Aren’t the samples very similar? Since the thinning interval is only 1 epoch."
> Preliminary experiments (not included in the paper) showed that using more samples and/or a larger thinning interval does not significantly improve performance. We will add a systematic evaluation of this matter to the paper; thank you for the suggestion!
>
> 8. "It would make the paper stronger if the authors are able to demonstrate the usefulness of detecting OoD in latent space through experiments."
> Thank you for this suggestion! We now added initial results for out-of-distribution detection in latent space to the paper (see Section 5.2) and plan to add further results (e.g., on more datasets).

---

### Author Response · Authors · 2019-11-15
**Paper revision with new experimental results**

We uploaded a revised version of our paper with a new set of experiments on a higher-dimensional benchmark, involving the SVHN and CIFAR10 datasets (as requested by reviewer #1); see Appendix A.
Please also note that we had previously uploaded a revision including additional experimental results on out-of-distribution detection in latent space (as requested by all three reviewers); see Section 5.2.
We would again like to thank all reviewers for their helpful suggestions to improve our paper!

---

### Decision · Program_Chairs · 2019-12-19

**Decision:**

Reject

**Comment:**

This paper tackles the problem of detection out-of-distribution (OoD) samples. The proposed solution is based on a Bayesian variational autoencoder. The authors show that information-theoretic measures applied on the posterior distribution over the decoder parameters can be used to detect OoD samples. The resulting approach is shown to outperform baselines in experiments conducted on three benchmarks (CIFAR-10 vs SVNH and two based on FashionMNIST).

Following the rebuttal, major concerns remained regarding the justification of the approach. The reason why relying on active learning principles should allow for OoD detection would need to be clarified, and the use of the effective sample size (ESS) would require a stronger motivation. Overall, although a theoretically-informed OoD strategy is indeed interesting and relevant, reviewers were not convinced by the provided theoretical justifications. I therefore recommend to reject this paper.